# Towards Scale-Invariant Graph-related Problem Solving by Iterative Homogeneous Graph Neural Networks

**Hao Tang**
Shanghai Jiao Tong University
tanghaosjtu@gmail.com

**Zhiao Huang**
UC San Diego
z2huang@eng.ucsd.edu

**Jiayuan Gu**
UC San Diego
jigu@eng.ucsd.edu

**Bao-Liang Lu**
Shanghai Jiao Tong University
bllu@sjtu.edu.cn

**Hao Su**
UC San Diego
haosu@eng.ucsd.edu

## Abstract

Current graph neural networks (GNNs) lack generalizability with respect to scales (graph sizes, graph diameters, edge weights, etc..) when solving many graph analysis problems. Taking the perspective of synthesizing graph theory programs, we propose several extensions to address the issue. First, inspired by the dependency of the iteration number of common graph theory algorithms on graph size, we learn to terminate the message passing process in GNNs adaptively according to the computation progress. Second, inspired by the fact that many graph theory algorithms are homogeneous with respect to graph weights, we introduce homogeneous transformation layers that are universal homogeneous function approximators, to convert ordinary GNNs to be homogeneous. Experimentally, we show that our GNN can be trained from small-scale graphs but generalize well to large-scale graphs for a number of basic graph theory problems. It also shows generalizability for applications of multi-body physical simulation and image-based navigation problems.

## 1 Introduction

Graph, as a powerful data representation, arises in many real-world applications [1, 2, 3, 4, 5, 6]. On the other hand, the flexibility of graphs, including the different representations of isomorphic graphs, the unlimited degree distributions [7, 8], and the *boundless graph scales* [9, 10], also presents many challenges to their analysis. Recently, Graph Neural Networks (GNNs) have attracted broad attention in solving graph analysis problems. They are permutation-invariant/equivariant by design and have shown superior performance on various graph-based applications [11, 12, 13, 14, 15].

However, investigation into *the generalizability of GNNs with respect to the graph scale* is still limited. Specifically, we are interested in GNNs that can learn from small graphs and perform well on new graphs of arbitrary scales. Existing GNNs [11, 12, 13, 15] are either ineffective or inefficient under this setting. In fact, even ignoring the optimization process of network training, the representation power of existing GNNs is yet too limited to achieve graph scale generalizability. There are at least two issues: 1) By using a pre-defined layer number [16, 17, 18], these GNNs are not able to approximate graph algorithms whose complexity depends on graph size (most graph algorithms in textbooks are of this kind). The reason is easy to see: For most GNNs, each node only uses information of the 1-hop neighborhoods to update features by message passing, and it is impossible for $k$-layer GNNs to send messages between nodes whose distance is larger than $k$. More formally,

Loukas [19] proves that GNNs, which fall within the message passing framework, lose a significant portion of their power for solving many graph problems when their width and depth are restricted; and 2) a not-so-obvious observation is that, the range of numbers to be encoded by the internal representation may deviate greatly for graphs of different scales. For example, if we train a GNN to solve the shortest path problem on small graphs of diameter $k$ with weight in the range of $[0, 1]$, the internal representation could only need to build the encoding for the path length within $[0, k]$; but if we test this GNN on a large graph of diameter $K \gg k$ with the same weight range, then it has to use and transform the encoding for $[0, K]$. The performance of classical neural network modules (e.g. the multilayer perceptron in GNNs) are usually highly degraded on those out-of-range inputs.

To address the pre-defined layer number issue, we take a program synthesis perspective, to design GNNs that have stronger representation power by mimicking the control flow of classical graph algorithms. Typical graph algorithm, such as Dijkstra's algorithm for shortest path computation, are iterative. They often consist of two sub-modules: an iteration body to solve the sub-problem (e.g., update the distance for the neighborhood of a node as in Dijkstra), and a termination condition to control the loop out of the iteration body. By adjusting the iteration numbers, an iterative algorithm can handle arbitrary large-scale problems. We, therefore, introduce our novel Iterative GNN (IterGNN) that equips ordinary GNN with an adaptive and differentiable stopping criterion to let GNN iterate by itself, as shown in Figure 1. Our stopping condition is adaptive to the inputs, supports arbitrarily large iteration numbers, and, interestingly, is able to be trained in an end-to-end fashion *without any direct supervision*.

We also give a partial solution to address the issue of out-of-range number encoding, if the underlying graph algorithm is in a specific hypothesis class. More concretely, the solutions to many graph problems, such as the shortest path problem and TSP problem, are homogeneous with respect to the input graph weights, i.e., the solution scales linearly with the magnitudes of the input weights. To build GNNs with representation power to approximate the solution to such graph problems, we further introduce the homogeneous inductive-bias. By assuming the message processing functions are homogeneous, the knowledge that neural networks learn at one scale can be generalized to different scales. We build HomoMLP and HomoGNN as powerful approximates of homogeneous functions over vectors and graphs, respectively.

We summarize our contributions as follows: **(1)** We propose IterGNN to approximate iterative algorithms, which avoids fixed computation steps in previous graph neural networks, and provides the potential for solving arbitrary large-scale problems. **(2)** The homogeneous prior is further introduced as a powerful inductive bias for solving many graph-related problems. **(3)** We prove the universal approximation theorem of HomoMLP for homogeneous functions and also prove the generalization error bounds of homogeneous neural networks under proper conditions. **(4)** In experiments, we demonstrate that our methods can generalize on various tasks and have outperformed baselines.

## 2 Related Work

**Graph Algorithm Learning.** Despite the success of GNNs (mostly come within the message passing framework [14, 15]) in many fields [13, 20, 11], few works have reported remarkable results on solving traditional graph-related problems, such as the shortest path problem, by neural networks, especially when the generalizability with regard to scales is taken into account. Neural Turing Machine [21, 22] first reported performance on solving the shortest path problem on small graphs using deep neural networks and Neural Logic Machine [23] solved the shortest path problem on graphs with limited diameters. Recently, [24], [25] and [26] achieved positive performance on graph algorithm learning on relatively large graphs using GNNs. However, [24, 25] require per-layer supervision to train, and models in [26] can not extend to large graph scales due to their bounded number of message passing steps. As far as we know, no previous work has solved the shortest path problem by neural networks on graphs of diameters larger than 100.

**Iterative Algorithm Approximation.** Inspired by the success of traditional iterative algorithms [27, 28], several works were proposed to incorporate the iterative architecture into neural networks for better generalizability [18, 29], more efficiency [30], or to support end-to-end training [16, 17]. However, none of them supports adaptive and unbounded iteration numbers and is therefore not applicable for approximating general iterative algorithms over graphs of any sizes.

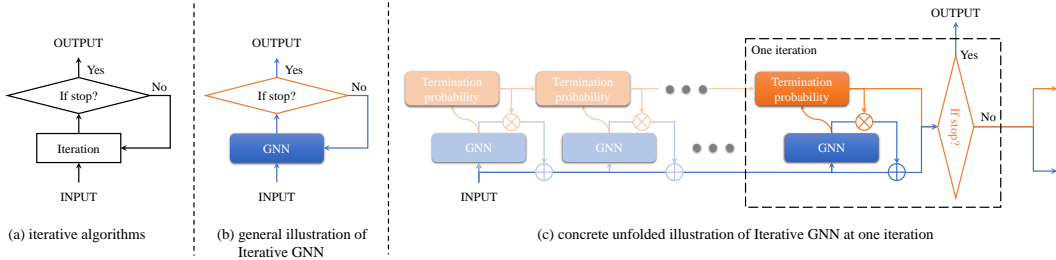

(a) iterative algorithms

(b) general illustration of Iterative GNN

(c) concrete unfolded illustration of Iterative GNN at one iteration

Figure 1: (a) The illustration of general iterative algorithms. The iteration body is repeated until the stopping criterion is satisfied. (b) Illustration of IterGNN as a combination of GNNs and iterative module. (c) A detailed illustration of Iterative GNN. It unfolds the computational flow of IterGNN. Other than the normal data flow (marked as blue), there is another control flow (marked as orange) that serves both as an adaptive stopping criterion and as a data flow controller.

**Differentiable Controlling Flows.** In recent years, multiple works have been proposed in the graph representation learning field that integrate controlling into neural networks to achieve flexible data-driven control. For example, DGCNN [31] implemented a differentiable sort operator (sort pooling) to build more powerful readout functions. Graph U-Net [32, 33] designed an adaptive pooling operator (TopK pooling) to support flexible data-driven pooling operations. All these methods achieved the differentiability by relaxing and multiplying the controlling signals with the neural networks' hidden representations. Inspired by their works, our method also differentiates the iterative algorithm by relaxing and multiplying the stopping criterion's output into neural networks' hidden representations.

**Adaptive Depth of Neural Networks.** The final formulation of our method is generally similar to the previous adaptive computation time algorithm (ACT) [34] for RNNs or spatially ACT [35, 36] for CNNs, however, with distinct motivations and formulation details. The numbers of iterations for ACT are usually small by design (e.g.the formulation of regularizations and halting distributions). Contrarily, Our method is designed to fundamentally improve the generalizability of GNNs w.r.t. scales by generalizing to much larger iteration numbers. Several improvements are proposed accordingly. The recent flow-based methods (e.g. the Graph Neural ODE [37]) are also potentially able to provide adaptive layer numbers. However, with no explicit iteration controller, they are not a straightforward solution to approximate iterative algorithms and to encode related inductive biases.

## 3 Backgrounds

**Graphs and graph scales.** Each graph $G := (V, E)$ consists of a set of nodes $V$ and a set of edges (pairs of nodes) $E$. To notate graphs with attributes, we use $\vec{x}_v$ for node attributes of node $v \in V$ and use $\vec{x}_e$ for edge attributes of edge $e \in E$. We consider three graph properties to quantify the graph scales, which are the number of nodes $N := |V|$, which is also called the graph size, the graph diameter $\delta_G := \max_{u,v \in V} d(u,v)$, and the scale of attributes' magnitudes $H := \max_{v \in V} ||\vec{x}_v|| + \max_{e \in E} ||\vec{x}_e||$. Here, $||\cdot||$ denotes an arbitrary norm of vectors and $d(u,v)$ denotes the length of the shortest path from node $u$ to node $v$, which is also called the distance between node $u$ and node $v$ for undirected graphs. We assume graph scales are unbounded but finite, and the aim is to generalize learned knowledge to graphs of arbitrary scales.

**Graph Neural Networks.** We describe a known class of GNNs that encompasses many state-of-art networks, including GCN [38], GAT [39], GIN [40], and Interaction Networks [4], among others. Networks with a global state [15] or utilizing multi-hop information per layer [41, 42, 43] can often be re-expressed within this class, as discussed in [19]. The class of GNNs generalizes the message-passing framework [14] to handle edge attributes. Each layer of it can be written as

$$\vec{h}_v^{(l+1)} = f_\theta^{(l)}(\vec{h}_v^{(l)}, \{\vec{x}_e : e \in \mathcal{N}_E(v)\}, \{\vec{h}_{v'}^{(l)} : v' \in \mathcal{N}_V(v)\}). \tag{1}$$

$\vec{h}_v^{(l)}$ is the node feature vector of node $v$ at layer $l$. $\mathcal{N}_V(v)$ and $\mathcal{N}_E(v)$ denote the sets of nodes and edges that are directly connected to node $v$ (i.e. its 1-hop neighborhood). $f_\theta^{(l)}$ is a parameterized function, which is usually composed of several multilayer perceptron modules and several aggregation functions (e.g. sum/max) in practice. Readers are referred to [11, 12, 13, 15] for thorough reviews.

# 4 Method

We propose Iterative GNN (IterGNN) and Homogeneous GNN (HomoGNN) to improve the generalizability of GNNs with respect to graph scales. IterGNN is first introduced, in Section 4.1, to enable adaptive and unbounded iterations of GNN layers so that the model can generalize to graphs of arbitrary scale. We further introduce HomoGNN, in Section 4.2, to partially solve the problem of out-of-range number encoding for graph-related problems. We finally describe PathGNN that improves the generalizability of GNNs for distance-related problems by improving the algorithm alignments [26] to the Bellman-Ford algorithm in Section 4.3.

## 4.1 Iterative module

The core of IterGNN is a differentiable iterative module. It executes the same GNN layer repeatedly until a learned stopping criterion is met. We present the pseudo-codes in Algorithm 1. At time step $k$, the iteration body $f$ updates the hidden states as $h^k = f(h^{k-1})$; the stopping criterion function $g$ then calculates a confidence score $c^k = g(h^k) \in [0, 1]$ to describe the probability of the iteration to terminate at this step. The module determines the number of iterations using a random process based on the confidence scores $c^k$. At each time step $k$, the random process has a probability of $c^k$ to terminate the iteration and to return the current hidden states $h^k$ as the output. The probability for the whole process to return $h^k$ is then $p^k = \left(\prod_{i=1}^{k-1}(1 - c^i)\right) c^k$, which is the

---

**Algorithm 1:** Iterative module. $g$ is the stopping criterion and $f$ is the iteration body

> **input:** initial feature $x$; stopping threshold $\epsilon$
> $k \leftarrow 1$
> $h^0 \leftarrow x$
> **while** $\prod_{i=1}^{k-1}(1 - c^i) > \epsilon$ **do**
> $\quad h^k \leftarrow f(h^{k-1})$
> $\quad c^k \leftarrow g(h^k)$
> $\quad k \leftarrow k + 1$
> **end while**
> **return** $h = \sum_{j=1}^{k-1} \left(\prod_{i=1}^{j-1}(1 - c^i)\right) c^j h^j$

---

product of the probabilities of continuing the iteration at steps from 1 to $k - 1$ and stopping at step $k$. However, the sampling procedure is not differentiable. Instead, we execute the iterative module until the "continue" probability $\prod_{i=1}^{k-1}(1 - c^i)$ is smaller than a threshold $\epsilon$ and return an expectation $h = \sum_{j=1}^{k} p^j h^j$ at the end. The gradient to the output $h$ thus can optimize the hidden states $h^k$ and the confidence scores $c^k$ jointly.

For example, assume $c^i = 0$ for $i < k$, $c^k = a$, $c^{k+1} = b$, and $(1-a)(1-b) < \epsilon$. If we follow the pre-defined random process, for steps before $k$, the iteration will not stop as $c^i = 0$ for $i < k$. For the step $k$, the process has a probability of $a$ to stop and output $h^k$; otherwise, the iteration will continue to the step $k + 1$. Similarly, at the step $k + 1$, the iteration has a probability of $b$ to stop and output $h^{k+1}$. We stop the iteration after step $k + 1$ as the "continue" probability $\prod_{i=1}^{k}(1 - c^i) = (1-a)(1-b)$ is negligible. The final output is the expectation of the output of the random process $h = ah^k + (1-a)bh^{k+1}$.

By setting $f$ and $g$ as GNNs, we obtain our novel IterGNN, as shown in Figure 1. The features are associated with nodes in the graph as $\{\vec{h}_v^{(k)} : v \in V\}$. GNN layers as described in Eq. 1 are adopted as the body function $f$ to update the node features iteratively $\{\vec{h}_v^{(k)} : v \in V\} = \text{GNN}(G, \{\vec{h}_v^{(k-1)} : v \in V\}, \{\vec{h}_e : e \in E\})$. We build the termination probability module as $g$ by integrating a readout function and an MLP. The readout function (e.g. max/mean pooling) summarizes all node features $\{\vec{h}_v^{(k)} : v \in V\}$ into a fixed-dimensional vector $\vec{h}^{(k)}$. The MLP predicts the confidence score as $c^k = \text{sigmoid}(\text{MLP}(\vec{h}^{(k)}))$. The sigmoid function is utilized to ensure the output of $g$ is between 0 and 1. With the help of our iterative module, IterGNN can adaptively adjust the number of iterations. Moreover, it can be trained without any supervision of the stopping condition.

Our iterative module can resemble the control flow of many classical graph algorithms since the iteration of most graph algorithms depends on the size of the graph. For example, Dijkstra's algorithm [27] has a loop to greedily propagate the shortest path from the source node. The number of iterations to run the loop depends linearly on the graph size. Ideally, we hope that our $f$ can learn the loop body and $g$ can stop the loop when all the nodes have been reached from the source. Interestingly, the experiment result shows such kind behavior. This structural level of the computation

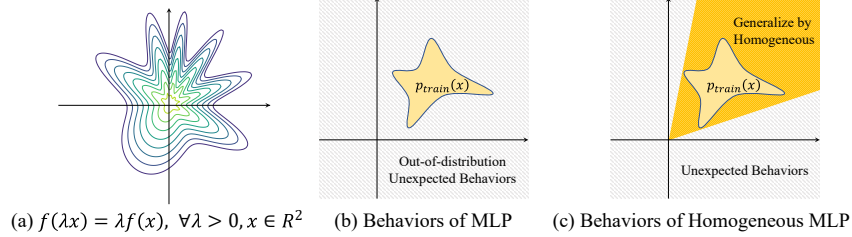

(a) $f(\lambda x) = \lambda f(x),\ \forall \lambda > 0, x \in R^2$    (b) Behaviors of MLP    (c) Behaviors of Homogeneous MLP

Figure 2: (a) An example of homogeneous functions. (b-c) Illustration of the improved generalizability by applying the homogeneous prior. Knowledge learned from the training samples not only can be generalized to samples of the same data distribution as ordinary neural networks, as shown in (b), but also can be generalized to samples of the scaled data distributions, as shown in (c).

allows superior generalizability, which agrees with the findings in [26] that improved algorithm alignment can increase network generalizability. In contrast, without a dynamic iterative module, previous GNNs have much inferior ability to generalize to larger graphs.

We state more details of IterGNN in Appendix, including the memory-efficient implementation, the theoretical analysis of representation powers, the node-wise iterative module to support unconnected graphs, and the decaying confidence mechanism to achieve much larger iteration numbers during inference in practice (by compensating the nonzero properties of the sigmoid function in $g$).

## 4.2 Homogeneous prior

The homogeneous prior is introduced to improve the generalizability of GNNs for out-of-range features/attributes. We first define the positive homogeneous property of a function:

**Definition 1** *A function $f$ over vectors is positive homogeneous iff $f(\lambda \vec{x}) = \lambda f(\vec{x})$ for all $\lambda > 0$.*

*A function $f$ over graphs is positive homogeneous iff for any graph $G = (V, E)$ with node attributes $\vec{x}_v$ and edge attributes $\vec{x}_e$, $f(G, \{\lambda \vec{x}_v : v \in V\}, \{\lambda \vec{x}_e : e \in E\}) = \lambda f(G, \{\vec{x}_v : v \in V\}, \{\vec{x}_e : e \in E\})$*

The solutions to most graph-related problems are positive homogeneous, such as the length of the shortest path, the maximum flow, graph radius, and the optimal distance in the traveling salesman problem.

The homogeneous prior tackles the problem of different magnitudes of features for generalization. As illustrated in Figure 2, by assuming functions as positive homogeneous, models can generalize knowledge to the scaled features/attributes of different magnitudes. For example, let us assume two datasets $D$ and $D_\lambda$ that are only different on magnitudes, which means $D_\lambda := \{\lambda x : x \in D\}$ and $\lambda > 0$. If the target function $f$ and the function $F_{\mathcal{A}}$ represented by neural networks $\mathcal{A}$ are both homogeneous, the prediction error on dataset $D_\lambda$ then scales linearly w.r.t. the scaling factor $\lambda$:

$$\sum_{x \in D_\lambda} ||f(x) - F_{\mathcal{A}}(x)|| = \sum_{x' \in D} ||f(\lambda x') - F_{\mathcal{A}}(\lambda x')|| = \lambda \sum_{x' \in D} ||f(x') - F_{\mathcal{A}}(x')||. \tag{2}$$

We design the family of GNNs that are homogeneous, named HomoGNN, as follows: simply remove all the bias terms in the multi-layer perceptron (MLP) used by ordinary GNNs, so that all affine transformations degenerate to linear transformations. Additionally, only homogeneous activation functions are allowed to be used. Note that ReLU is a homogeneous activation function. The original MLP used in ordinary GNNs become HomoMLP in HomoGNNs afterward.

### 4.2.1 Theoretical analysis of HomoGNN and HomoMLP

We provide theoretical proofs showing that, if the target function is homogeneous, low generalization errors and low training errors are both achievable using the pre-defined homogeneous neural networks under proper conditions. We first formalize the generalization error bounds of homogeneous neural networks on approximating homogeneous functions under some assumptions, by extending the previous example to more general cases. To show that low training errors are achievable, we further prove that HomoMLP is a universal approximator of the homogeneous functions under proper

conditions, based on the universal approximation theorem for width-bounded ReLU networks [44]. We present propositions stating that HomoGNN and HomoMLP can only represent homogeneous functions, along with the proofs for all theorems, in the Appendix.

Let training samples $D_m = \{x_1, x_2, \cdots x_m\}$ be independently sampled from the distribution $\mathcal{D}_x$, then if we scale the training samples with the scaling factor $\lambda \in \mathbb{R}^+$ which is independently sampled from the distribution $\mathcal{D}_\lambda$, we get a "scaled" distribution $\mathcal{D}_x^\lambda$, which has a density function $P_{\mathcal{D}_x^\lambda}(z) := \int_\lambda \int_x \delta(\lambda x = z) P_{\mathcal{D}_\lambda}(\lambda) P_{\mathcal{D}_x}(x) \, \mathrm{d}x \, \mathrm{d}\lambda$. The following theorem bounds the generalization error bounds on $\mathcal{D}_x^\lambda$:

**Theorem 1** (Generalization error bounds of homogeneous neural networks with independent scaling assumption). *For any positive homogeneous functions function $f$ and neural network $F_\mathcal{A}$, let $\beta$ bounds the generalization errors on the training distribution $D_x$, i.e., $\mathbb{E}_{x \sim \mathcal{D}_x} |f(x) - \mathcal{F}_\mathcal{A}(x)| \le \frac{1}{m} \sum_{i=1}^m |f(x_i) - F_\mathcal{A}(x_i)| + \beta$, then the generalization errors on the scaled distributions $\mathcal{D}_x^\lambda$ scale linearly with the expectation of scales $\mathbb{E}_{\mathcal{D}_\lambda}[\lambda]$:*

$$\mathbb{E}_{x \sim \mathcal{D}_x^\lambda} |f(x) - F_\mathcal{A}(x)| = \mathbb{E}_{\mathcal{D}_\lambda}[\lambda] \mathbb{E}_{x \sim \mathcal{D}_x} |f(x) - F_\mathcal{A}(x)| \le \mathbb{E}_{\mathcal{D}_\lambda}[\lambda] (\frac{1}{m} \sum_{i=1}^m |f(x_i) - F_\mathcal{A}(x_i)| + \beta) \quad (3)$$

**Theorem 2** (Universal approximation theorem for width-bounded HomoMLP). *For any positive-homogeneous Lebesgue-integrable function $f : \mathbb{X} \mapsto \mathbb{R}$, where $\mathbb{X}$ is a Lebesgue-measurable compact subset of $\mathbb{R}^n$, and for any $\epsilon > 0$, there exists a finite-layer HomoMLP $\mathcal{A}'$ with width $d_m \le 2(n+4)$, which represents the function $F_{\mathcal{A}'}$ such that $\int_\mathbb{X} |f(x) - F_{\mathcal{A}'}(x)| \, \mathrm{d}x < \epsilon$.*

### 4.3 Path graph neural networks

We design PathGNN to imitate one iteration of the classical Bellman-Ford algorithm. It inherits the generalizability of the Bellman-Ford algorithm and the flexibility of the neural networks. Specifically, the Bellman-Ford algorithm performs the operation $dist_i = \min(dist_i, \min_{j \in \mathcal{N}(i)}(dist_j + w_{ji}))$ iteratively to solve the shortest path problem, where $dist_i$ is the current estimated distance from the source node to the node $i$, and $w_{ji}$ denotes the weight of the edge from node $j$ to node $i$. If we consider $dist_i$ as node features and $w_{ij}$ as edge features, one iteration of the Bellman-Ford algorithm can be exactly reproduced by GNN layers as described in Eq. 1:

$$\vec{h}_i = \min(\vec{h}_i, \min_{j \in \mathcal{N}(i)} (\vec{h}_j + \vec{x}_{ji})) \equiv -\max(-\vec{h}_i, \max_{j \in \mathcal{N}(i)} (-\vec{h}_j - \vec{x}_{ji})).$$

To achieve more flexibilities for solving problems other than the shortest path problem, we integrate neural network modules, such as MLPs to update features or the classical attentional-pooling to aggregate features, while building the PathGNN layers. A typical variant of PathGNN is as follows:

$$\alpha_{ji} = \mathrm{softmax}(\{\mathrm{MLP}_1(\vec{h}_j; \vec{h}_i; \vec{x}_{ji}) \text{ for } j \in \mathcal{N}(i)\});$$
$$\vec{h}_i' = \sum_{j \in \mathcal{N}(i)} \alpha_{ji} \mathrm{MLP}_2(\vec{h}_j; \vec{h}_i; \vec{x}_{ji}); \quad \vec{h}_i = \max(\vec{h}_i, \vec{h}_i'),$$

We state the detailed formulation and variations of PathGNN layers in the Appendix.

## 5 Experiments

Our experimental evaluation aims to study the following empirical questions: **(1)** Will our proposals, the PathGNN layer, the homogeneous prior, and the iterative module, improve the generalizability of GNNs with respect to graph scales that are the number of nodes, the diameter of graphs, and the magnitude of attributes? **(2)** Will our iterative module adaptively change the iteration numbers and consequently learn an interpretable stopping criterion in practice? **(3)** Can our proposals improve the performance of general graph-based reasoning tasks such as those in physical simulation, image-based navigation, and reinforcement learning?

**Graph theory problems and tasks.** We consider three graph theory problems, i.e., shortest path, component counting, and Traveling Salesman Problem (TSP), to evaluate models' generalizability

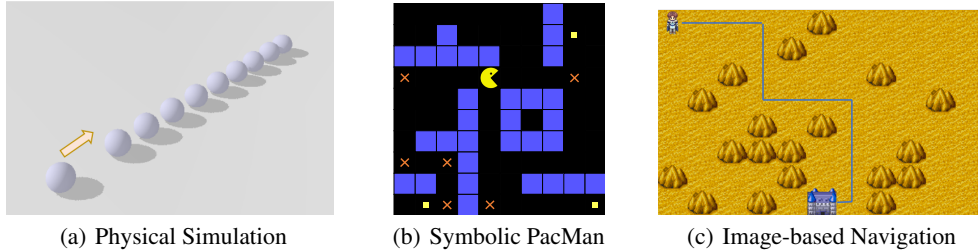

| (a) Physical Simulation | (b) Symbolic PacMan | (c) Image-based Navigation |

Figure 3: Figure (a) shows a set of Newton's balls in the physical simulator. The yellow arrow is the moving direction of the first ball. Figure (b) shows our symbolic PacMan environment. Figure (c) illustrates our image-based navigation task in a RPG-game environment.

w.r.t. graph scales. We build a benchmark by combining multiple graph generators, including Erdos-Renyi (ER), K-Nearest-Neighborhoods graphs (KNN), planar graphs (PL), and lobster graphs (Lob), so that the generated graphs can have more diverse properties. We further apply our proposals to three graph-related reasoning tasks, i.e., physical simulation, symbolic Pacman, and image-based navigation, as illustrated in Figure 3. The generation processes and the properties of datasets are listed in the Appendix.

**Models and baselines.** Previous problems and tasks can be formulated as graph regression/classification problems. We thus construct models and baselines following the common practice [15, 31, 40]. We stack 30 GCN [38]/GAT [39] layers to build the baseline models. GIN [40] is not enlisted since 30-layer GINs do not converge in most of our preliminary experiments. Our "Path" model stacks 30 PathGNN layers. Our "Homo-Path" model replaces GNNs and MLPs in the "Path" model with HomoGNNs and HomoMLPs. Our "Iter-Path" model adopts the iterative module to control the iteration number of the GNN layer in the "Path" model. The final "Iter-Homo-Path" integrates all proposals together. Details are in the Appendix.

**Training Details.** We utilize the default hyper-parameters to train models. We generate 10000 samples for training, 1000 samples for validation, and 1000 samples for testing. The only two tunable hyper-parameter in our experiment is the epoch number (10 choices) and the formulation of PathGNN layers (3 choices). Validation datasets are used to tune them. More details are listed in the Appendix.

## 5.1 Solving graph theory problems

**Generalize w.r.t. graph sizes and graph diameters.** We present the generalization performance for all three graph theory problems in Table 1. Models are trained on graphs of sizes within $[4, 34]$ and are evaluated on graphs of larger sizes such as 100 (for shortest path and TSP) and 500 (for component counting so that the diameters of components are large enough). The relative loss metric is defined as $|y - \hat{y}|/|y|$, given a label $y$ and a prediction $\hat{y}$. The results demonstrate that each of our proposals improves the generalizability on almost all problems. Exceptions happen on graphs generated by ER. It is because the diameters of those graphs are 2 with high probability even though the graph sizes are large. Our final model, Iter-Homo-Path, which integrates all proposals, performs much better than the baselines such as GCN and GAT. The performance on graphs generated by KNN and PL further supports the analysis. The concrete results are presented in the Appendix due to space limitations. We also evaluated a deeper Path model, i.e., with 100 layers, on the weighted shortest path problem (Lob). The generalization performance (relative loss $\approx 0.13$) became even worse.

We then explore models' generalizability on much larger graphs on the shortest path problem using Lob to generate graphs with larger diameters. As shown in Table 2, our model achieves a 100% success rate of identifying the shortest paths on graphs with as large as 5000 nodes even though it is trained on graphs of sizes within $[4, 34]$. As claimed, the iterative module is necessary for generalizing to graphs of much larger sizes and diameters due to the message passing nature of GNNs. The iterative module successfully improves the performance from $\sim 60\%$ to 100% on graphs of sizes $\geq 500$.

**Ablation studies and comparison.** We conduct ablation studies to exhibit the benefits of our proposals using the unweighted shortest path problem on lobster graphs with 1000 nodes in Table 3.

Table 1: Generalization performance on graph algorithm learning and graph-related reasoning. Models are trained on graphs of smaller sizes (e.g., within $[4, 34]$ or $\leq 10 \times 10$) and are tested on graphs of larger sizes (e.g., 50, 100, 500, $16 \times 16$ or $33 \times 33$). The metric for the shortest path and TSP is the relative loss. The metric for component counting is accuracy. The metric for physical simulation is the mean square error. The metric for image-based navigation is the success rate.

| | Graph Theory Problems | | | | | Graph-related Reasoning | | | |
| | Shortest Path | | Component Cnt. | | TSP | Physical sim. | | Image-based Navi. | |
| Models | ER | Lob | ER | Lob | 2D | 50 | 100 | $16 \times 16$ | $33 \times 33$ |
|---|---|---|---|---|---|---|---|---|---|
| GCN [38] | 0.1937 | 0.44 | 0.0% | 0.0% | 0.52 | 42.18 | 121.14 | 34.2% | 28.9% |
| GAT [39] | 0.1731 | 0.28 | 24.4 % | 0.0% | 0.18 | >1e4 | >1e4 | 56.7% | 44.5% |
| Path (ours) | **0.0003** | 0.29 | 82.3% | 77.2% | 0.16 | 20.24 | 27.67 | 85.6% | 65.1% |
| Homo-Path (ours) | 0.0008 | 0.27 | 91.9% | 83.9% | 0.14 | 20.48 | 21.45 | 87.8% | 69.3% |
| Iter-Path (ours) | 0.0005 | 0.09 | 86.7% | 96.1% | 0.08 | **0.13** | 1.68 | 89.4% | 78.6% |
| Iter-Homo-Path (ours) | 0.0007 | **0.02** | **99.6%** | **97.5%** | **0.07** | **0.07** | 2.01 | **98.8%** | **91.7%** |

Table 2: Generalization performance on the shortest path problem with lobster graphs. During training, node numbers are within $[4, 34]$ for unweighted problems (whose metric is the success rate), and edge weights are within $[0.5, 1.5)$ for weighted problems (whose metric is the relative loss).

| Generalize | w.r.t. sizes and diameters - unweighted | | | | | w.r.t. magnitudes - weighted | | | |
| | 20 | 100 | 500 | 1000 | 5000 | $[0.5, 1.5)$ | $[1, 3)$ | $[2, 6)$ | $[8, 24)$ |
|---|---|---|---|---|---|---|---|---|---|
| GCN [38] | 66.6 | 25.7 | 5.5 | 2.4 | 0.4 | 0.31 | 0.37 | 0.49 | 0.56 |
| GAT [39] | **100.0** | 42.7 | 10.5 | 5.3 | 0.9 | 0.13 | 0.29 | 0.49 | 0.55 |
| Path (ours) | **100.0** | 62.9 | 20.1 | 10.3 | 1.6 | 0.06 | 0.22 | 0.44 | 0.54 |
| Homo-Path (ours) | **100.0** | 58.3 | 53.7 | 50.2 | 1.6 | 0.03 | **0.03** | **0.03** | **0.03** |
| Iter-Homo-Path (ours) | **100.0** | **100.0** | **100.0** | **100.0** | **100.0** | **0.01** | 0.04 | 0.06 | 0.08 |

The models are built by replacing each proposal in our best Iter-Homo-Path model with other possible substitutes in the literature. For the iterative module, other than the simplest paradigm utilized in Homo-Path that stacks GNN layers sequentially, we also compare it with the ACT algorithm [34] and the fixed-depth weight-sharing paradigm [18, 16], resulting in the "ACT-Homo-Path" and "Shared-Homo-Path" models. The ACT algorithm provides adaptive but usually short iterations of layers (see Figure 4 and Appendix). The weight-sharing paradigm iterates modules for predefined times and assumes that the predefined iteration number is large enough. We set its iteration number to the largest graph size in the dataset. Homo-Path and ACT-Homo-Path perform much worse than Iter/Shared-Homo-Path because of the limited representation powers of shallow GNNs. Shared-Homo-Path performs worse than our Iter-Homo-Path, possibly because of the accumulated errors after unnecessary iterations. For the homogeneous prior, we build "Iter-Path" by simply removing the homogeneous prior. It performs much worse than Iter-Homo-Path because of the poor performance of MLPs on out-of-distribution features. For PathGNN, we build "Iter-Homo-GCN" and "Iter-Homo-GAT" by replacing PathGNN with GCN and GAT. Their bad performance verifies the benefits of better algorithm alignments [26].

**Generalize w.r.t. magnitudes of attributes.** We evaluate the generalizability of models w.r.t. magnitudes of attributes using the weighted shortest path length problem, as shown in Table 2. The edge weights are randomly sampled from $[0.5, 1.5)$ during training and are sampled from $[1, 3)$, $[2, 6)$,

| Iter-Homo-Path | |
|---|---|
| **100.0** | |
| Homo-Path | Iter-Path |
| 53.7 | 48.9 |
| ACT-Homo-Path | Iter-Homo-GAT |
| 52.7 | 2.9 |
| Shared-Homo-Path | Iter-Homo-GCN |
| 91.7 | 1.4 |

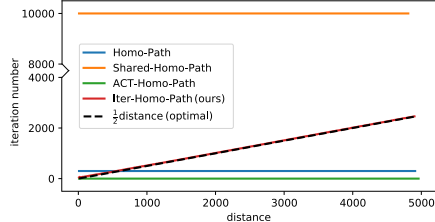

Table 3: Ablation studies of generalization performance for the shortest path problem on lobster graphs with 1000 nodes. Metric is the success rate.

Figure 4: The iteration numbers of GNN layers w.r.t. the distances from the source node to the target node for the shortest path problem.

and $[8, 24)$ during evaluations. The distributions of node numbers remain the same. As claimed, our models successfully generalize to graphs of different magnitudes with far better performance than baselines. Notably, the Homo-Path model even achieves the same performance (relative loss $\approx 0.03$) for all scales of magnitudes, which experientially supports Theorem 1. The Iter-Homo-Path model performs slightly worse because the sigmoid function in the iterative module is not homogeneous.

**Interpreting stopping criterion learned by the iterative module.** We show that our Iter-Homo-Path model learned the optimal stopping criterion for the unweighted shortest path problem in Figure 4. Typically, to accurately predict the shortest path of lengths $d$ on undirected graphs, the iteration number of GNN layers is at least $d/2$ due to the message passing nature of GNNs (see Appendix for details). Our iterative module learned such optimal stopping criterion. The Iter-Homo-Path model adaptively increases the iteration numbers w.r.t. the distances and, moreover, stops timely when the information is enough.

## 5.2 General reasoning tasks

**Physical simulation.** We evaluate the generalizability of our models by predicting the moving patterns between objects in a physical simulator. We consider an environment called *Newton's ball*: all balls with the same mass lie on a friction-free pathway. With the ball at one end moving towards others, our model needs to predict the motion of the balls of both ends at the next time step. The metric is the mean squared error. Models are trained in worlds with $[4, 34)$ balls and are tested in worlds with 100 balls. As shown in Table 1, the Iter-Homo-Path model and the Iter-Path model significantly outperform others, demonstrating the advantages of our iterative module for improving generalizability w.r.t. scales. The homogeneous prior is not as beneficial since the target functions are not homogeneous.

**Symbolic PacMan.** To show that our iterative module can improve reinforcement learning, we construct a symbolic PacMan environment with similar rules to the PacMan in Atari [45]. The environment contains a map with dots and walls. The agent needs to figure out a policy to quickly "eat" all dots while avoiding walls on the map to maximize the return. We abstract the observations as graphs using the landmark [46]. We adopt Double Q learning [47] to train the policy. Unlike original Atari PacMan, our environment is more challenging because we randomly sample the layout of maps for each episode, and we test models in environments with different numbers of dots and walls. The agent cannot just remember one policy to be successful but needs to learn to do planning according to the current observation. The metric is the success rate of eating dots. Our IterGNN (97.5%) performs much better than baselines, CNN (91.5%) and PointNet [48] (29.0%). Our IterGNN also shows remarkable generalizability among different environment settings. For example, even though the models are trained in environments with 10 dots and 8 walls, our IterGNN achieves a 94.0% success rate in environments with 10 dots and 15 walls and 93.4% in environments with 8 walls and 20 dots. The tables that list the generalization performance of IterGNN, GCN, and PointNet in 30 different settings of environments are presented in the Appendix to save space.

**Image-based navigation.** We show the benefits of the differentiability of a generalizable reasoning module using the image-based navigation task. The model needs to plan the shortest route from the source to target on 2D images with obstacles. However, the properties of obstacles are not given as a prior, and the model must discover them based on image patterns during training. We simplify the task by defining each pixel as obstacles merely according to its own pixel values. As stated in Table 1, our Iter-Homo-Path model successfully solves the task. The model achieves success rates larger than 90% for finding the shortest paths on images of size $16 \times 16$, and $33 \times 33$, while it is only trained on images of size $\leq 10 \times 10$. All of our proposals help improve generalizability.

## 6 Conclusion

We propose an iterative module and the homogeneous prior to improve the generalizability of GNNs w.r.t. graph scales. Experiments show that our proposals do improve the generalizability for solving multiple graph-related problems and tasks.

# 7 Acknowledgements

H. Tang and B. -L. Lu were supported in part by the National Key Research and Development Program of China (2017YFB1002501), the National Natural Science Foundation of China (61673266 and 61976135), SJTU Trans-med Awards Research (WF540162605), the Fundamental Research Funds for the Central Universities, and the 111 Project. H. Su, Z. Huang, and J. Gu were supported by the NSF grant IIS-1764078. We specially thank Zhizuo Zhang, Zhiwei Jia, and Chutong Yang for the useful discussions, and Wei-Long Zheng, Bingyu Shen and Yuming Zhao for reviewing the paper prior to submission.

# 8 Broader Impact

Our methods provide general tools to improve the generalizability of GNNs with respect to scales. This work can thus be applied to many applications of GNNs, such as natural language processing, traffic prediction, and recommendation systems. They have many potential positive impact in the society. For example, better traffic prediction enables shorter traffic time for all vehicles, which could help protect the environment. Improved recommendation system could promote the transition of information for more productivity and more fairness. Moreover, by improving the generalizability with respect to scales, models can be trained on graphs of much smaller scales than reality. It reduces the cost of collecting and storing large datasets with large samples, which can then alleviate the risks of violating privacy and of harming the environment. On the other hand, this work may also have negative consequences. Improving techniques in the field of natural language processing can help monitor and collect personal information of each individual. Stronger recommendation system can also hurt the fairness as different information targeted to different groups of people.

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
