[Supplementary Material]

# Supplementary Material of Towards Scale-Invariant Graph-related Problem Solving by Iterative Homogeneous Graph Neural Networks

## A    Organization of the Appendices

In the supplementary materials, we aim to answer the following questions: **(1)** How powerful is our iterative module on approximating the iterative algorithms? **(2)** Are low generalization errors achievable when using homogeneous neural networks to approximate the homogeneous functions? What is the generation error bound? **(3)** Are low training errors achievable when using HomoMLP to approximate the homogeneous functions? Is HomoMLP a universal approximator of positive homogeneous functions? **(4)** Is the iterative module harmful to the standard generalizability in practice? What is its performance on graph-classification benchmarks? **(5)** What are the experimental setups? How are the models built?

To answer the question **(1)**, we present the theoretical analysis of the representation power of our iterative module in Section B.1. To answer the question **(2)**, we prove the generalization error bounds of homogeneous neural networks on approximating homogeneous functions with independent scaling assumptions in Section B.2.1. A concrete bound based on the PAC-Bayesian framework is also presented in Lemma B.2.1. To answer the question **(3)**, we present and prove the universal approximation theorem on approximating the homogeneous functions for both general HomoMLP and width-bounded HomoMLP in Section B.2.3. To answer question **(4)**, we show that our IterGIN model, which wraps each layer of the state-of-art GIN [1] model with our iterative module, achieves competitive performance to GIN, in Section E.3. To answer question **(5)**, we describe all omitted details of the experimental setups in Section D.

Moreover, we provide the index of contents following the same order as they appear in the main body of the paper, as follows:

- Details of IterGNN, such as the memory-efficient implementation in Section C.1.3, the theoretical analysis of representation powers in Section B.1, the node-wise iterative module to support unconnected graphs in Section C.1.1, and the decaying confidence mechanism to achieve much larger iteration numbers during inference in practice in Section C.1.2.
- Theoretical analysis of homogeneous neural networks, including the proof of Theorem 1 in Section B.2.1, the proof of Theorem 2 in Section B.2.3, the propositions stating that HomoGNN and HomoMLP can only represent homogeneous functions in Section B.2.2.
- The formulation details of PathGNN layers in Section C.2.
- The generation processes and the properties of datasets in Section D.
- The details of models and baselines in Section D.4.
- The training details in Section D.5.
- The ACT algorithm usually learns small iteration numbers in Section E.1.2.
- The minimum depth of GNNs to accurately predict the shortest path of length $l$ is $l/2$ in Section E.1.2.

- The generalization performance of IterGNN, GCN, and PointNet on the symbolic Pacman task in environments with different number of dots and different number of walls in Section E.2.

In general, we provide the theoretical analysis of our proposals in Section B. We describe the detailed formulations of our proposals in Section C. The omitted experimental setups are all listed in Section D and the omitted experimental results are presented in Section E. At last, we also state more background knowledges of graph neural networks (GNNs) in Section F.

## B    Theoretical analysis

We present the theoretical analysis of our proposals in this section. Main results include

- The representation powers of our iterative module.
    - Our iterative module is a universal approximator of the iterative algorithms, with oracles to reproduce the body function and the condition function in the iterative algorithms. (Theorem B.1.1)
    - Under some more practical assumptions, we show that our iterative module can achieve adaptive and unbounded iteration numbers depending on the graph sizes using GNNs. (Proposition B.1.1)
- Generation error bounds of homogeneous neural networks.
    - We prove that the generation error bounds of homogeneous neural networks on approximating the homogeneous functions scale linearly with the expectation of the scales/magnitudes under the independent scaling assumption. (Theorem 1)
    - We provide a concrete generation error bounds for homogeneous neural networks by integrating Theorem 1 and a specific generation error bounds with classical i.i.d. assumptions in the PAC-Bayesian framework (Eq.7 in [2]). (Lemma B.2.1)
- The homogeneous properties of HomoMLP and of HomoGNN.
    - We prove that HomoMLP and HomoGNN can only represent homogeneous functions. (Proposition B.2.1 and Proposition B.2.2)
- The universal approximation theorems of homogeneous functions for HomoMLP.
    - We prove that HomoMLP is a universal approximator of homogeneous functions. (Theorem B.2.2)
    - We prove that width-bounded HomoMLP is also a universal approximator of homogeneous functions. (Theorem 2)

### B.1    Representation powers of iterative module

We first state the intuition that our iterative module as described in the main body can approximate any iterative algorithms as defined in Algorithm 1, as long as the *body* and *condition* functions are available or can be perfectly reproduced by neural networks.

---
**Algorithm 1:** Iterative algorithm

   **input**  initial feature $x$
   $k \leftarrow 1$
   $h^0 \leftarrow x$
   **while** not $condition(h^k)$ **do**
      $h^k \leftarrow body(h^{k-1})$
      $k \leftarrow k + 1$
   **end while**
   **return** $h = h^k$

---

More formally, we build an ideal class of models, named as Iter-Oracle, by combining our iterative module with the oracles $\mathcal{F}_\theta$ that can perfectly reproduce the *body* function and the *condition* function,

which means there exist $\theta'$ and $\theta''$ such that for all $x$, $\mathcal{F}_{\theta'}(x) = body(x)$ and $\mathcal{F}_{\theta''}(x) = condition(x)$. We can then show the representation power of our iterative module using the following theorem:

**Theorem B.1.1.** *For any iterative algorithm, iter-alg, defined as in Algorithm 1, for any initial feature $x$, and for any $\epsilon > 0$, there exist an Iter-Oracle model $\mathcal{A}$, which represents the function $\mathcal{F}_{\mathcal{A}}$, satisfying*

$$||iter\text{-}alg(x) - \mathcal{F}_{\mathcal{A}}(x)|| < \epsilon. \tag{1}$$

We prove it by construction. We build the function $f$ in our iterative module using $\mathcal{F}_{\theta'}$ such that $\mathcal{F}_{\theta'}(x) = body(x)$ for all $x$. The function $g$ in our iterative module is built as sigmoid($\alpha(\mathcal{F}_{\theta''}(x) - 0.5)$), where $\mathcal{F}_{\theta''}(x) = condition(x)$ for all $x$ and we utilize similar rules to the python language for type conversion, which means $condition(x)$ outputs 1 if the condition is satisfied and outputs 0 otherwise. By setting $\alpha \to +\infty$, we have

$$c^k \to \begin{cases} 1 & \text{if } condition(h^k) \\ 0 & \text{if not } condition(h^k). \end{cases}$$

Therefore, $\sum_{j=1}^{\infty} c^j h^j \prod_{i=1}^{j-1}(1 - c^i) \to h^k$, where $k$ is the time step that $condition(h^k)$ is firstly satisfied. More formally, assume $\Lambda$ bounds the norm of features $h^j$ for the specific iterative algorithm $iter\text{-}alg$, for the specific initial feature $x$, and for any $j$, we can set $\alpha$ as

$$\alpha > 2\ln\left(\frac{(k+1)\Lambda}{\epsilon} - 1\right) \text{ and } \alpha > -2\ln\left(\left(1 + \frac{\epsilon}{(k+1)\Lambda}\right)^{-\frac{1}{k}} - 1\right), \tag{2}$$

so that Eq. 1 is satisfied, since for $j < k$,

$$\left\| c^j h^j \prod_{i=1}^{j-1}(1 - c^i) - 0 \right\| < \left\| c^j h^j \right\| < \frac{1}{1 + e^{\frac{\alpha}{2}}}\Lambda = \frac{\epsilon}{k+1}, \tag{3}$$

for $j = k$,

$$\left\| c^k h^k \prod_{i=1}^{k-1}(1 - c^i) - h^k \right\| < \left(\left(\frac{1}{1 + e^{-\frac{\alpha}{2}}}\right)^k - 1\right)\Lambda = \frac{\epsilon}{k+1}, \tag{4}$$

for $j > k$,

$$\left\| \sum_{j=k+1}^{\infty} c^j h^j \prod_{i=1}^{j-1}(1 - c^i) - 0 \right\| < (1 - c^k)\Lambda < \frac{1}{1 + e^{\frac{\alpha}{2}}}\Lambda = \frac{\epsilon}{k+1}. \tag{5}$$

Together, we have

$$\left\| \sum_{j=1}^{\infty} c^j h^j \prod_{i=1}^{j-1}(1 - c^i) - h^k \right\| \leq \sum_{j=1}^{k-1}\left\| c^j h^j \prod_{i=1}^{j-1}(1 - c^i) - 0 \right\| + \tag{6}$$

$$\left\| c^k h^k \prod_{i=1}^{k-1}(1 - c^i) - h^k \right\| + \tag{7}$$

$$\left\| \sum_{j=k+1}^{\infty} c^j h^j \prod_{i=1}^{j-1}(1 - c^i) - 0 \right\| \tag{8}$$

$$< (k+1)\frac{\epsilon}{k+1} = \epsilon \quad \square$$

We then derive a more practical proposition of IterGNN, based on Theorem B.1.1, stating the intuition that IterGNN can achieve adaptive and unbounded iteration numbers:

**Proposition B.1.1.** *Under the assumptions that IterGNN can calculate graph sizes $N$ with no error and the multilayer perceptron used by IterGNN is a universal approximator of continuous functions on compact subsets of $\mathbb{R}^n$ ($n \geq 1$) (i.e. the universal approximation theorem), there exist IterGNNs whose iteration numbers are constant, linear, polynomial or exponential functions of the graph sizes.*

The proofs are simple. Let $g'$ be the function that maps the graph sizes $N$ to iteration numbers $k$. we can then build GNNs as $f$ to calculate the graph sizes $N$ and the number of current time step $j$, and build $g$ as sigmoid($\alpha(0.5 - |g'(N) - j|)$). The $\alpha$ can be set similarly to the previous proof except for one scalar that compensates the approximation errors of neural networks. More formally, assume $\epsilon'$ bounds the error of predicting $g'(N)$ and $j$ using neural networks $\mathcal{A}$, which means for any input, there exists neural networks that represent functions $\mathcal{F}_g$ and $\mathcal{F}_j$ satisfying $|g'(N) - \mathcal{F}_g| < \epsilon'$ and $|j - \mathcal{F}_j| < \epsilon'$. We can then set $\alpha$ as

$$\alpha > \frac{2}{1 - 4\epsilon'} \ln\left(\frac{(k+1)\Lambda}{\epsilon} - 1\right) \text{ and } \alpha > -\frac{2}{1 - 4\epsilon'} \ln\left(\left(1 + \frac{\epsilon}{(k+1)\Lambda}\right)^{-\frac{1}{k}} - 1\right), \quad (9)$$

so that Proposition B.1.1 is satisfied. Note that it is easy to build GNNs to exactly calculate the graph sizes $N$ and the number of current time step $j$. Given the universal approximation theorem of MLP, the stopping condition function $g$ can also be easily approximated by MLPs. Our iterative module is thus able to achieve adaptive and unbounded iteration numbers using neural networks.

## B.2  Homogeneous prior

We formalize the generalization error bounds of homogeneous neural networks on approximating homogeneous functions under proper conditions, by extending the example in the main body to more general cases, in Section B.2.1. To make sure functions represented by neural networks are homogeneous, we also prove that HomoGNN and HomoMLP can only represent homogeneous functions, in Section B.2.2. To show that low training errors are achievable, we further analyze the representation powers of HomoMLP and demonstrate that it is a universal approximator of homogeneous functions under some assumptions, based on the universal approximation theorem for width-bounded ReLU networks [3], in Section B.2.3.

### B.2.1  Proof of Theorem 1: generalization error bounds of homogeneous neural networks

Extending the example in the main body to more general cases, we present the out-of-distribution generalization error bounds of homogeneous neural networks on approximating homogeneous functions under the assumption of independent scaling of magnitudes during inference:

Let training samples $D_m = \{x_1, x_2, \cdots x_m\}$ be independently sampled from the distribution $\mathcal{D}_x$, then if we scale the training samples with the scaling factor $\lambda \in \mathbb{R}^+$ which is independently sampled from the distribution $\mathcal{D}_\lambda$, we get a "scaled" distribution $\mathcal{D}_x^\lambda$, which has a density function $P_{\mathcal{D}_x^\lambda}(z) := \int_\lambda \int_x \delta(\lambda x = z) P_{\mathcal{D}_\lambda}(\lambda) P_{\mathcal{D}_x}(x) \, \mathrm{d}x \, \mathrm{d}\lambda$. The following theorem bounds the generalization error bounds on $\mathcal{D}_x^\lambda$:

**Theorem 1.** (Generalization error bounds of homogeneous neural networks with independent scaling assumption). *For any positive homogeneous functions function $f$ and neural network $F_\mathcal{A}$, let $\beta$ bounds the generalization errors on the training distribution $D_x$ , i.e., $\mathbb{E}_{x \sim \mathcal{D}_x}|f(x) - \mathcal{F}_\mathcal{A}(x)| \leq \frac{1}{m}\sum_{i=1}^m |f(x_i) - F_\mathcal{A}(x_i)| + \beta$, then the generalization errors on the scaled distributions $\mathcal{D}_x^\lambda$ scale linearly with the expectation of scales $\mathbb{E}_{\mathcal{D}_\lambda}[\lambda]$:*

$$\mathbb{E}_{x \sim \mathcal{D}_x^\lambda}|f(x) - F_\mathcal{A}(x)| = \mathbb{E}_{\mathcal{D}_\lambda}[\lambda]\mathbb{E}_{x \sim \mathcal{D}_x}|f(x) - F_\mathcal{A}(x)| \leq \mathbb{E}_{\mathcal{D}_\lambda}[\lambda](\frac{1}{m}\sum_{i=1}^m |f(x_i) - F_\mathcal{A}(x_i)| + \beta) \ (10)$$

The proof is as simple as re-expressing the formulas:

$$\mathbb{E}_{z \sim \mathcal{D}_x^{\lambda}} |f(z) - F_{\mathcal{A}}(z)| = \int_{\lambda} \int_x P_{\mathcal{D}_{\lambda}}(\lambda) P_{\mathcal{D}_x}(x) |f(\lambda x) - F_{\mathcal{A}}(\lambda x)| \, dx \, d\lambda \quad (11)$$

$$= \int_{\lambda} P_{\mathcal{D}_{\lambda}}(\lambda) \lambda \, d\lambda \int_x P_{\mathcal{D}_x}(x) |f(x) - F_{\mathcal{A}}(x)| \, dx \quad (12)$$

$$= \mathbb{E}_{\mathcal{D}_{\lambda}}[\lambda] \mathbb{E}_{x \sim \mathcal{D}_x} |f(x) - F_{\mathcal{A}}(x)| \quad (13)$$

$$\leq \mathbb{E}_{\mathcal{D}_{\lambda}}[\lambda] (\frac{1}{m} \sum_{i=1}^m |f(x_i) - F_{\mathcal{A}}(x_i)| + \beta) \quad \square$$

Theorem 1 can be considered as a meta-bound and can thus be integrated with any specific generalization error bounds with classical i.i.d. assumptions to create a concrete generation error bounds of homogeneous neural networks on approximating homogeneous functions with independent scaling assumptions. For example, when integrated a generation error bounds in the PAC-Bayesian framework (Eq.7 in [2]), we obtain the following lemma: Let $f_w$ be any predictor learned from training data. We consider a distribution $\mathcal{Q}$ over predictors with weights of the form $w + v$, where $w$ is a single predictor learned from the training set, and $v$ is a random variable.

**Lemma B.2.1.** *Assume all hypothesis $h$ and $f_{w+v}$ for any $v$ are positive homogeneous functions, as defined in Definition 1. Then, given a "prior" distribution $P$ over the hypothesis that is independent of the training data, with probability at least $1 - \delta$ over the draw of the training data, the expected error of $f_{w+v}$ on the scaled distribution $\mathcal{D}_x^{\lambda}$ can be bounded as follows*

$$\mathbb{E}_{v} \left[ \mathbb{E}_{x \sim \mathcal{D}_x^{\lambda}} \left[ |f(x) - f_{w+v}(x)| \right] \right] \leq \mathbb{E}_{\mathcal{D}_{\lambda}} [\lambda] \left( \mathbb{E}_{v} \left[ \frac{1}{m} \sum_{i=1}^m |f(x_i) - f_{w+v}(x_i)| \right] + 4 \sqrt{\frac{1}{m} \left( KL(w + v || P) + \ln \frac{2m}{\delta} \right)} \right).$$

### B.2.2 HomoMLP and HomoGNN are homogeneous functions

We present that HomoGNN and HomoMLP can only represent homogeneous functions:

**Proposition B.2.1.** *For any input $x$, we have HomoMLP($\lambda x$) = $\lambda$HomoMLP($x$) for all $\lambda > 0$.*

**Proposition B.2.2.** *For any graph $G = (V, E)$ with node attributes $\vec{x}_v$ and edge attributes $\vec{x}_e$, we have for all $\lambda > 0$,*

*HomoGNN($G, \{\lambda \vec{x}_v : v \in V\}, \{\lambda \vec{x}_e : e \in E\}$) = $\lambda$HomoGNN($G, \{\vec{x}_v : v \in V\}, \{\vec{x}_e : e \in E\}$). (14)*

Both propositions are derived from the following lemma:

**Lemma B.2.2.** *Compositions of homogeneous functions are homogeneous functions.*

We compose functions by taking the outputs of functions as the input of other functions. The inputs of functions are either the outputs of other functions or the initial features $x$. For example, we can compose functions $f, g, h$ as $h(f(x), g(x), x)$. If $f, g, h$ are all homogeneous functions, we have for all $x$ and all $\lambda > 0$,

$$h(f(\lambda x), g(\lambda x), \lambda x) = \lambda h(f(x), g(x), x). \quad (15)$$

More formally, we can prove the lemma by induction. We denote the composition of a set of functions $\{f_1, f_2, \cdots, f_n\}$ as $O(\{f_1, f_2, \cdots, f_n\})$. Note that there are multiple ways to compose $n$ functions. Here, $O(\{f_1, f_2, \cdots, f_n\})$ just denotes one specific way of composition, and we can use $O'(\{f_1, f_2, \cdots, f_n\})$ to denote another. We want to prove that the composition of homogeneous functions is still a homogeneous function, which means for all $n > 0$, for all $\lambda > 0$, and for all possible ways of compositions $O$, $O(\{f_1, f_2, \cdots, f_n\})(\lambda x) = \lambda O(\{f_1, f_2, \cdots, f_n\})(x)$.

The base case: $n = 1$. The composition of a single function $O(\{f_1\})$ is itself $f_1$. Therefore, $O(\{f_1\})$ is homogeneous by definition as $O(\{f_1\})(\lambda x) = f_1(\lambda x) = \lambda O(\{f_1\})(x)$.

Assume the composition of the $k$ functions is homogeneous, for any composition of $k + 1$ functions $O(\{f_1, f_2, \cdots, f_{k+1}\})$, let $f_{k+1}$ be the last

function of the compositions, which means $O(\{f_1, f_2, \cdots, f_{k+1}\})(x) :=$ $f_{k+1}(O'(\{f_1, f_2, \cdots, f_k\})(x), O''(\{f_1, f_2, \cdots, f_k\})(x), O'''(\{f_1, f_2, \cdots, f_k\})(x), \cdots)$, it's easy to see that, according to the definition of homogeneous functions,

$$O(\{f_1, f_2, \cdots, f_{k+1}\})(\lambda x) = \lambda O(\{f_1, f_2, \cdots, f_{k+1}\})(x). \quad \square$$

Therefore, we prove the Lemma B.2.2. Proposition B.2.1 and Proposition B.2.1 can all be considered as specializations of this lemma.

### B.2.3 Proof of Theorem 2: representation powers of HomoMLP

We first introduce a class of neural networks (defined in Eq.16) that can be proved as a universal approximator of homogeneous functions (Theorem B.2.1) and then show that our HomoMLP is as powerful as this class of neural networks to prove that our HomoMLP is a universal approximator of homogeneous functions (Theorem B.2.2). Furthermore, we prove the universal approximation theorem for width-bounded HomoMLP (Theorem 2).

We construct a class of neural networks as the universal approximator of positive homogeneous functions, as defined in the main body, as follows:

$$G_{\text{MLP}}(x) = |x|\mathcal{F}_{\text{MLP}}(\frac{x}{|x|}), \tag{16}$$

where $|\cdot|$ denotes the L1 norm, the MLP denotes the classical multilayer perceptrons with positive homogeneous activation functions and $\mathcal{F}_{\text{MLP}}$ is the function represented by the specific MLP.

**Proposition B.2.3.** *For any input $x$, we have $G_{MLP}(\lambda x) = \lambda G_{MLP}(x)$.*

This proposition can be easily proved by re-expressing the formulas:

$$G_{\text{MLP}}(\lambda x) = |\lambda x|\mathcal{F}_{\text{MLP}}(\frac{\lambda x}{|\lambda x|}) = \lambda |x|\mathcal{F}_{\text{MLP}}(\frac{x}{|x|}) = \lambda G_{\text{MLP}}(x) \quad \square$$

We show that it is a universal approximator of homogeneous functions: Let $\mathbb{X}$ be a compact subset of $\mathbb{R}^m$ and $C(\mathbb{X})$ denotes the space of real-valued continuous functions on $\mathbb{X}$.

**Theorem B.2.1.** (Univeral approximation theorem for $G_{\text{MLP}}$.) *Given any $\epsilon > 0$ and any function $f \in C(\mathbb{X})$, there exist a finite-layer feed-forward neural networks $\mathcal{A}$ with positive homogeneous activation functions such that for all $x \in \mathbb{X}$, we have $|G_{\mathcal{A}}(x) - f(x)| = \left| |x|\mathcal{F}_{\mathcal{A}}\left(\frac{x}{|x|}\right) - f(x) \right| < \epsilon$.*

The prove is as simple as applying the universal approximation theorem of MLPs [4] and applying the definition of the homogeneous functions. In detail, as $\mathbb{X}$ is a compact subset, the magnitudes of inputs $x$ is bounded. We use $M$ denote the bound, which means $|x| \leq M$ for all $x \in \mathbb{X}$. According to the universal approximation theorem of MLPs [4], there exists a finite-layer feed-forward layer $\mathcal{A}$ with ReLU as activation functions such that $\left| \mathcal{F}_{\mathcal{A}}\left(\frac{x}{|x|}\right) - f\left(\frac{x}{|x|}\right) \right| < \frac{\epsilon}{M}$ for all $x \in \mathbb{X}$. Then, according to definition of homogeneous functions, we have

$$|G_{\mathcal{A}}(x) - f(x)| = \left| |x|\mathcal{F}_{\mathcal{A}}\left(\frac{x}{|x|}\right) - |x|f\left(\frac{x}{|x|}\right) \right| < |x|\frac{\epsilon}{M} < \epsilon \tag{17}$$

Note that ReLU is positive homogeneous. Therefore, we finish the proof of Theorem B.2.1. $\square$

We then prove that HomoMLP is a universal approximator of homogeneous functions: Let $\mathbb{X}$ be a compact subset of $\mathbb{R}^m$ and $C(\mathbb{X})$ denotes the space of real-valued continuous functions on $\mathbb{X}$.

**Theorem B.2.2.** (Universal approximation theorem for HomoMLP.) *Given any $\epsilon > 0$ and any function $f \in C(\mathbb{X})$, there exist a finite-layer HomoMLP $\mathcal{A}'$, which represents the function $F_{\mathcal{A}'}$, such that for all $x \in \mathbb{X}$, we have $|F_{\mathcal{A}'}(x) - f(x)| < \epsilon$.*

We prove it based on Theorem B.2.1 by construction. In detail, according to Theorem B.2.1, there exists a finite-layer MLP $\mathcal{A}$ such that for all $x \in \mathbb{X}$, $|G_{\mathcal{A}}(x) - f(x)| = \left| |x|\mathcal{F}_{\mathcal{A}}\left(\frac{x}{|x|}\right) - f(x) \right| < \epsilon$.

Without loss of generality, we assume $\mathcal{A}$ as a two-layer ReLU feed-forward neural networks, which means $\mathcal{F}_{\mathcal{A}}(x) = W^2\text{ReLU}(W^1 x + b^1) + b^2$, where $W^1 \in \mathbb{R}^{n \times m}, W^2 \in \mathbb{R}^{1 \times n}$ are weight matrices and $b^1 \in \mathbb{R}^n, b^2 \in \mathbb{R}$ are biases. The function $G_{\mathcal{A}}$ can then be expressed as

$$G_{\mathcal{A}}(x) = |x|\left(W^2\text{ReLU}\left(W^1\frac{x}{|x|} + b^1\right) + b^2\right) = W^2\text{ReLU}\left(W^1 x + b^1|x|\right) + b^2|x|. \quad (18)$$

Therefore, we just need to prove that the L1 norm $|\cdot|$ can be exactly calculated by HomoMLP to show that HomoMLP is a universal approximator homogeneous functions. Typically, we construct a two-layer HomoMLP as follows

$$\mathbf{1}^T\text{ReLU}\left(\begin{bmatrix}\mathbf{I}\\-\mathbf{I}\end{bmatrix}x\right) \equiv |x|, \text{for all } x \in \mathbb{R}^m, \quad (19)$$

where $\mathbf{1}$ is a vertical vector containing $2m$ elements whose values are all one, and $\mathbf{I}$ denotes the identity matrix of size $m \times m$. Together with Eq. 18, we show that $G_{\mathcal{A}}(x)$ is a specification of HomoMLP, denoted as $\mathcal{A}'$, as follows:

$$G_{\mathcal{A}}(x) = W^2\text{ReLU}\left(W^1 x + b^1|x|\right) + b^2|x| \quad (20)$$

$$= W^2\text{ReLU}\left(W^1 x + b^1\mathbf{1}^T\text{ReLU}\left(\begin{bmatrix}\mathbf{I}\\-\mathbf{I}\end{bmatrix}x\right)\right) + b^2\mathbf{1}^T\text{ReLU}\left(\begin{bmatrix}\mathbf{I}\\-\mathbf{I}\end{bmatrix}x\right) \quad (21)$$

$$= \begin{bmatrix}W^2 & b^2\end{bmatrix}\text{ReLU}\left(\begin{bmatrix}W^1 + b^1 & -W^1 + b^1\\\mathbf{1}^T & \mathbf{1}^T\end{bmatrix}\text{ReLU}\left(\begin{bmatrix}\mathbf{I}\\-\mathbf{I}\end{bmatrix}x\right)\right) \quad (22)$$

$$= F_{\mathcal{A}'}(x) \quad (23)$$

Here, we use $\mathbf{1}$ to denote a vertical vector containing $m$ elements whose values are all one, and $\mathbf{I}$ still denotes the identity matrix of size $m \times m$. The summation of the weight matrix $W \in \mathbb{R}^{n \times m}$ and the bias $b \in \mathbb{R}^n$ outputs a new weight matrix $W_b$ such that $W_b[i,j] = W[i,j] + b[i]$ for all $i = 1, 2, \cdots, n$ and $j = 1, 2, \cdots, m$, where $W[i,j]$ is the element in the $i$th row and the $j$th column of matrix $W$ and $b[i]$ is the $i$th element of vertex $b$. Consequently, we build a three-layer HomoMLP $\mathcal{A}'$ with ReLU as activation functions such that for all $x \in \mathbb{X}$,

$$|F_{\mathcal{A}'}(x) - f(x)| = |G_{\mathcal{A}}(x) - f(x)| < \epsilon. \quad \square$$

Applying similar techniques to previous proofs, we can further derive the universal approximation theorem for width-bounded MLP. Theorem B.2.3 and Theorem 2 are proved to show that width-bounded $G_{\text{MLP}}$ and width-bounded HomoMLP are universal approximators of homogeneous functions.

**Theorem B.2.3.** (Universal approximation theorem for width-bounded $G_{\text{MLP}}$). *For any positive-homogeneous Lebesgue-integrable function $f : \mathbb{X} \mapsto \mathbb{R}$, where $\mathbb{X}$ is a Lebesgue-measurable compact subset of $\mathbb{R}^n$, and for any $\epsilon > 0$, there exists a finite-layer feed-forward neural networks $\mathcal{A}$ with positive homogeneous activation functions and with width $d_m \leq n + 4$, such that*

$$\int_{\mathbb{X}}|f(x) - G_{\mathcal{A}}(x)|\,\mathrm{d}x := \int_{\mathbb{X}}\left||x|\mathcal{F}_{\mathcal{A}}\left(\frac{x}{|x|}\right) - f(x)\right|\,\mathrm{d}x < \epsilon. \quad (24)$$

The proof is very similar to the proof of Theorem B.2.1. In detail, as $\mathbb{X}$ is a compact subset, the magnitudes of inputs $x$ is bounded. We use $M$ denote the bound, which means $|x| \leq M$ for all $x \in \mathbb{X}$. According to the universal approximation theorem of width-bounded MLPs [3], there exists a finite-layer feed-forward layer $\mathcal{A}$ with ReLU as activation functions and with width $d_m < n + 4$, such that

$$\int_{\mathbb{X}}\left|\mathcal{F}_{\mathcal{A}}\left(\frac{x}{|x|}\right) - f\left(\frac{x}{|x|}\right)\right|\,\mathrm{d}x < \frac{\epsilon}{M}. \quad (25)$$

Then, according to the definition of homogeneous functions, we have

$$\int_{\mathbb{X}}|f(x) - G_{\mathcal{A}}(x)|\,\mathrm{d}x = \int_{\mathbb{X}}\left||x|\mathcal{F}_{\mathcal{A}}\left(\frac{x}{|x|}\right) - |x|f\left(\frac{x}{|x|}\right)\right|\,\mathrm{d}x < M\frac{\epsilon}{M} = \epsilon. \quad (26)$$

Note that ReLU is positive homogeneous. Therefore, we finish the proof of Theorem B.2.3. $\square$

**Theorem 2.** (Universal approximation theorem for width-bounded HomoMLP with reasonable assumption). *For any positive-homogeneous Lebesgue-integrable function $f : \mathbb{X} \mapsto \mathbb{R}$, where $\mathbb{X}$ is a Lebesgue-measurable compact subset of $\mathbb{R}^n$, and for any $\epsilon > 0$, there exists a finite-layer HomoMLP $\mathcal{A}'$ with width $d_m \leq 2(n+4)$, which represents the function $F_{\mathcal{A}'}$ such that $\int_{\mathbb{X}} |f(x) - F_{\mathcal{A}'}(x)| \, dx < \epsilon$.*

The proof is very similar to the proof of Theorem B.2.2. In detail, according to Theorem B.2.3, there exists a finite-layer MLP $\mathcal{A}$ such that,

$$\int_{\mathbb{X}} |f(x) - G_{\mathcal{A}}(x)| \, dx = \int_{\mathbb{X}} \left| |x| \, \mathcal{F}_{\mathcal{A}} \left( \frac{x}{|x|} \right) - f(x) \right| dx < \epsilon. \tag{27}$$

Assume the MLP $\mathcal{A}$ is formulated as

$$\mathcal{F}_{\mathcal{A}}(x) = W^K \mathrm{ReLU} \left( W^{K-1} \mathrm{ReLU} \left( \cdots \mathrm{ReLU} \left( W^1 x + b^1 \right) \cdots \right) + b^{K-1} \right) + b^K, \tag{28}$$

where $K$ is the layer number, $W^1, W^2, \cdots, W^K$ are weight matrices, and $b^1, b^2, \cdots, b^K$ are biases. We can then re-express $G_{\mathcal{A}}(x)$, using the definition in Eq. 16, as follows

$$G_{\mathcal{A}}(x) = W^K \mathrm{ReLU} \left( W^{K-1} \mathrm{ReLU} \left( \cdots \mathrm{ReLU} \left( W^1 x + b^1 |x| \right) + \cdots \right) + b^{K-1} |x| \right) + b^K |x|. \tag{29}$$

Together with Eq. 19, we show that $G_{\mathcal{A}}(x)$ is a specification of HomoMLP, denoted as $\mathcal{A}'$, as follows:

$$
\begin{aligned}
G_{\mathcal{A}}(x) &= W^K \mathrm{ReLU} \left( W^{K-1} \mathrm{ReLU} \left( \cdots \mathrm{ReLU} \left( W^1 x + b^1 |x| \right) + \cdots \right) + b^{K-1} |x| \right) + b^K |x| \\[2mm]
&= \begin{bmatrix} W^K & b^K \end{bmatrix} \mathrm{ReLU} \left( \begin{bmatrix} W^{K-1} & b^{K-1} \\ \mathbf{0}^T & \mathbf{1}^T \end{bmatrix} \mathrm{ReLU}(\cdots \begin{bmatrix} W^2 & b^2 \\ \mathbf{0}^T & \mathbf{1}^T \end{bmatrix} \mathrm{ReLU}( \right. \\[2mm]
&\qquad \left. \begin{bmatrix} W^1 + b^1 & -W^1 + b^1 \\ \mathbf{1}^T & \mathbf{1}^T \end{bmatrix} \mathrm{ReLU}( \begin{bmatrix} \mathbf{I} \\ -\mathbf{I} \end{bmatrix} x)) \cdots )) \right. \\[2mm]
&= F_{\mathcal{A}'}(x).
\end{aligned}
$$

Here, we also use $\mathbf{1}$ to denote vectors full of ones, $\mathbf{0}$ to denote vectors full of zeros, and $\mathbf{I}$ to denote the identity matrix of size $m \times m$. The summation of the weight matrix $W \in \mathbb{R}^{n \times m}$ and the bias $b \in \mathbb{R}^n$ outputs a new weight matrix $W_b$ such that $W_b[i,j] = W[i,j] + b[i]$ for all $i = 1, 2, \cdots, n$ and $j = 1, 2, \cdots, m$, where $W[i,j]$ is the element in the $i$th row and the $j$th column of matrix $W$ and $b[i]$ is the $i$th element of vertex $b$. Consequently, we build a $(K+1)$-layer HomoMLP $\mathcal{A}'$ with ReLU as activation functions and with width $d_m \leq \max(2n, n+5)$ such that,

$$\int_{\mathbb{X}} |f(x) - F_{\mathcal{A}'}(x)| \, dx = \int_{\mathbb{X}} |f(x) - G_{\mathcal{A}}(x)| \, dx < \epsilon. \quad \square$$

## C  Method

We present more details about our iterative module in Section C.1, such as the memory-efficient implementation in Section C.1.3, the node-wise iterative module to support unconnected graphs in Section C.1.1, and the decaying confidence mechanism to achieve much larger iteration numbers during inference in practice in Section C.1.2. We describe how to formulate homogeneous neural network modules in Section C.3. We show the formulation of PathGNN layers in detail in Section C.2. There are three variants of PathGNN, each of which corresponds to different degrees of flexibilities of approximating functions. At last, we present the random initialization technique for solving the component counting problem using GNNs and discuss its motivations in Section C.4.

### C.1  Iterative module

We propose IterGNN to break the limitation of fixed-depth graph neural networks so that models can generalize to graphs of arbitrary scales. The core of IterGNN is a differentiable iterative module as described in the main body. We present more details about its formulations and implementations in this subsection. Its representation powers are analyzed in Section B.1.

### C.1.1 Node-wise iterative module

In the main body, we assume all nodes within the same graph share the same scale, so that we predict a single confidence score for the whole graph at each time step while building IterGNNs. However, for problems like connected component counting, the graphs can have multiple components of different scales. We can then utilize node-wise IterGNNs to achieve better performance. It is equivalent to apply our iteration module as described in the main body to each node (instead of to each graph). The models can thus learn to iterate for different times for nodes/components of different scales.

In detail, we also set the iteration body function $f$ and the stopping criterion function $g$ as GNNs. The body function $f$ still update the node features iteratively $\{\vec{h}_v^{(k+1)} : v \in V\} = \text{GNN}(G, \{\vec{h}_v^{(k)} : v \in V\}, \{\vec{h}_e : e \in E\})$. On the other hand, We build the termination probability module $g$ by node-wise embedding modules and node-wise prediction modules. Typically, we apply the same MLP to features of each node to predict the confidence score of each node $c_v^k = \text{sigmoid}(\text{MLP}(\vec{h}_v^{(k)}))$ at time step $k$. We similarly take the expectation of node features as the output, however with different distributions for different nodes: $\vec{h}_v = \sum_{k=1}^{\infty} c_v^k \prod_{i=1}^{k-1} (1 - c_v^i) \vec{h}_v^k$.

### C.1.2 Decaying confidence mechanism

Although the vanilla IterGNN, as described in the main body, theoretically supports infinite iteration numbers, models can hardly generalize to much larger iteration numbers during inference in practice. In detail, we utilize the sigmoid function to ensure that confidence scores are between 0 and 1. However, the sigmoid function can't predict zero confidence scores to continue iterations forever. Alternatively, the models will predict small confidence scores when they are not confident enough to terminate at the current time step. As a result, the models will still work well given the IID assumption, but can't generalize well when much larger iteration numbers are needed than those met during training. For example, while solving the shortest path problem, 0.05 is a sufficiently small confidence score during training, because no iteration number larger than 30 is necessary and $0.95^{30}$ is still quite larger than 0. However, such models can not generalize to graphs with node numbers larger than 300, since $0.95^{300} \to 0$ and the models will terminate before time step 300 in any case. During our preliminary experiments, the vanilla IterGNN can not iterate for more than 100 times.

The key challenge is the difference between iteration numbers during training and inference. We then introduce a simple decaying mechanism to achieve larger iteration numbers during inference. The improved algorithm is shown in Algorithm 2. The termination probabilities will manually decay/decrease by $\lambda$ at each time step. The final formulation of IterGNN can then generalize to iterate for 2500 times during inference in our experiments.

We compare the choices of decaying ratios as $0.99, 0.999, 0.9999$ in our preliminary experiments and fix it to $0.9999$ afterwards in all experiments.

---

**Algorithm 2:** IterGNN with decay. $g$ is the stopping criterion and $f$ is the iteration body

---

**Input:** initial feature $x$; stop threshold $\epsilon$; decay constant $\lambda$;
$k \leftarrow 1$
$h^0 \leftarrow x$
**while** $\lambda^k \prod_{i=1}^{k-1} (1 - c^i) > \epsilon$ **do**
   $h^k \leftarrow f(h^{k-1})$
   $c^k \leftarrow g(h^k)$
   $k \leftarrow k + 1$
**end while**
**return** $h = \lambda^k \sum_{j=1}^{k} c^j h^j \prod_{i=1}^{j-1} (1 - c^i)$

---

### C.1.3 Efficient train and inference by IterGNNs

The advantages of IterGNN are not only limited to improving the generalizability w.r.t. scales. For example, IterGNN also promotes the standard generalizability because of its better algorithm alignment [5] to iterative algorithms. In this subsection, we show that IterGNN moreover improves efficiencies for both training and inference. Briefly speaking, the improved generalizability w.r.t.

scales enables training on smaller graphs while still achieving satisfied performance on larger graphs. The cost of computations and the cost of memories are therefore decreased on those smaller graphs during training. During inference, we implement a memory-efficient algorithm by expressing the same logic with different formulas.

In detail, training GNNs take memories whose sizes scale at least linearly (in general quadratically) with respect to the graph sizes. In our preliminary experiments, 11GB GPU memories are not enough to train 30-layer GNNs on graphs of node numbers larger than 100 when the batch size is 32. It is therefore either infeasible or super-inefficient to train 500-layer GNNs directly on graphs with 1000 nodes, to meet the IID assumptions. We couldn't even fit such models within 32GB CPU memories for training. On the other hand, with the help of IterGNN, fewer iterations and smaller graphs are needed for training to achieve satisfying performance on larger graphs.

During inference, thanks to the equivalent formulations of IterGNNs as depicted in Algorithm 3, we don't need to store the node features at all times steps until the final output of our iterative module, as done in Algorithm 2. In practice, we can calculate the final output step by step as follows

---

**Algorithm 3:** IterGNN's efficient inference

    **Input:** initial feature $x$; stop threshold $\epsilon$; decay constant $\lambda$;
    $k \leftarrow 1$; $h^0 \leftarrow x$; $\bar{c} \leftarrow 1$; $\tilde{h} \leftarrow \vec{0}$;
    **while** $\bar{c} > \epsilon$ **do**
        $h^k \leftarrow f(h^{k-1})$; $c^k \leftarrow g(h^k)$;
        $\tilde{h} \leftarrow \lambda\tilde{h} + \bar{c}c^k h^k$
        $\bar{c} \leftarrow \lambda(1 - c^k)\bar{c}$;
        $k \leftarrow k + 1$
    **end while**
    **return** $h = \tilde{h}$

---

Note that this memory-efficient algorithm is only applicable during inference, since the node features at each time step must be stored during training to calculate the gradients during backward passes.

## C.2 Path Graph Neural Networks

As stated in [6], the performance of GNNs, especially their generalizability and zero-shot transferability, is largely influenced by the relational inductive biases. Xu [5] further formalized the relational inductive biases as sample efficiencies from algorithm alignments. For solving path-related graph problems such as shortest path, a classical algorithm is the Bellman-Fold algorithm. Therefore, to achieve more effective and generalizable solvers for path-related graph problems, we design several specializations of GN blocks, as described in [6] and in Section F.1, by exploiting the inductive biases of the Bellman-Fold algorithms. The notations are also presented in Section F.1.

Our first observation of the Bellman-Fold algorithm is that it directly utilizes the input attributes such as the edge weights and the source/target node identifications at each iteration. We further observe that the input graph attributes of classical graph-related problems are all informative, well defined and also well represented as discrete one-hot encodings or simple real numbers (e.g. edge weights). Therefore, we directly concatenate the input node attributes with the hidden node attributes as the new node attributes before fed into our Graph Networks (GN) blocks. The models then don't need to extract and later embed the input graph attributes into the hidden representations in each GN block.

Our second observation is that Bellman-Fold algorithm can be perfectly represented by Graph Networks as stated in Algorithm 4. Typically, for each iteration of the Bellman-Fold algorithm, the message module sums up the sender node's attributes (i.e. distance) with the edge weights, the node-level aggregation module then selects the minimum of all edge messages, and finally the attributes of the central node are updated if the new message (/distance) is smaller. The other modules of GN blocks are either identity functions or irrelevant. Therefore, to imitate the Bellman-Fold algorithm by GN blocks, we utilize max pooling for both aggregation and update modules to imitate the min poolings in the Bellman-Fold algorithm. The edge message module is MLP, similarly to most GNN variants. The resulting module is then equal to replacing the MPNN [7]'s aggregation and update module with max poolings. Therefore, we call it MPNN-Max as in [8]. The concrete formulas

---

**Algorithm 4:** The Bellman-Fold algorithm

---

**Input:** node attributes $V = \{\mathbf{v}_i, i = 1, 2, \cdots, N_v\}$, edge attributes $E = \{(w_k, s_k, r_k), k = 1, 2, \cdots, N_e\}$, and the source node *source*.
**Output:** The shortest path length from the source node to others, $distance = \{dist_i, i = 1, 2, \cdots, N_v\}$.

Initialize the intermediate node attributes $V' = \{dist_i, i = 1, 2, \cdots, N_v\}$ as
$$dist_i = \begin{cases} 0 & if. \ i = source \\ \infty & o.w. \end{cases}.$$
**for** i=1 **to** $N_v - 1$ **do**
   **for** $(w_k, s_k, r_k)$ **in** $E$ **do**
      $\mathbf{e}'_k = dist_{s_k} + w_k$   ◁ edge message module
   **end for**
   **for** j=1 **to** $N_v$ **do**
      $\bar{\mathbf{e}}_j = \min(\{\mathbf{e}'_k : r_k = j\})$   ◁ aggregation module
      $dist_j = \min(dist_j, \bar{\mathbf{e}}_j)$   ◁ update module
   **end for**
**end for**

---

---

**Algorithm 5:** One step of PathGNN

---

**Input:** graph $G = (V, E)$
**Output:** updated graph $G' = (V', E)$

**for** $(w_k, s_k, r_k)$ **in** $E$ **do**
   $\tilde{\mathbf{e}}_k = MLP(\mathbf{v}_{s_k}, \mathbf{v}_{r_k}, \mathbf{e}_k)$
   $score_k = MLP'(\mathbf{v}_{s_k}, \mathbf{v}_{r_k}, \mathbf{e}_k)$
   $\mathbf{e}'_k = (score_k, \tilde{\mathbf{e}}_k)$   ◁ edge message module
**end for**
**for** j=1 **to** $N_v$ **do**
   $\bar{\mathbf{e}}_j = \text{attention}(\{\mathbf{e}'_k : r_k = j\})$   ◁ aggregation module
   $\mathbf{v}'_j = \max(\mathbf{v}_j, \bar{\mathbf{e}}_j)$   ◁ update module
**end for**

---

are as follows

$$\begin{aligned}
\mathbf{e}'_k &= MLP(\mathbf{v}_{s_k}, \mathbf{v}_{r_k}, \mathbf{e}_k) \\
\bar{\mathbf{e}}_j &= \max(\{\mathbf{e}'_k : r_k = j\}) \\
\mathbf{v}'_j &= \max(\mathbf{v}_j, \bar{\mathbf{e}}_j)
\end{aligned}$$

Moreover, we notice that the Bellman-Ford algorithm is only designed for solving the shortest path problem. Many path-related graph problems can not be solved by it. Therefore, we further relax the max pooling to attentional poolings to increase the models' flexibility while still maintaining the ability to approximate min pooling in a sample efficient way. Typically, we propose the PathGNN by replacing the aggregation module with attentional pooling. The detailed algorithm is stated in Algorithm 5, where the global attributes are omitted due to their irrelevance.

---

**Algorithm 6:** Attention Pooling in GNNs

---

**Input:** set of messages $\{\mathbf{e}'_k = (score_k, \tilde{\mathbf{e}}_k)\}$
**Output:** aggregated messages $\bar{\mathbf{e}}_j$

$\alpha = softmax(score)$
$\bar{\mathbf{e}}_j = \sum_k \alpha_k \tilde{\mathbf{e}}_k$

---

---

**Algorithm 7:** One step of PathGNN-sim

---

**Input:** graph $G = (V, E)$
**Output:** updated graph $G' = (V', E)$

**for** $(w_k, s_k, r_k)$ **in** $E$ **do**
    $\tilde{\mathbf{e}}_k = MLP(\mathbf{v}_{s_k}, \mathbf{e}_k)$
    $score_k = MLP'(\mathbf{v}_{s_k}, \mathbf{v}_{r_k}, \mathbf{e}_k)$
    $\mathbf{e}'_k = (score_k, \tilde{\mathbf{e}}_k)$   ⊲ edge message module
**end for**
**for** j=1 to $N_v$ **do**
    $\bar{\mathbf{e}}_j = \text{attention}(\{\mathbf{e}'_k : r_k = j\})$   ⊲ aggregation module
    $\mathbf{v}'_j = \max(\mathbf{v}_j, \bar{\mathbf{e}}_j)$   ⊲ update module
**end for**

---

Another variant of PathGNN is also designed by exploiting a less significant inductive bias of the Bellman-Fold algorithm. Specifically, we observe that only the sender node's attributes and the edge attributes are useful in the message module while approximating the Bellman-Fold algorithm. Therefore, we only feed those attributes into the message module of our new PathGNN variant, PathGNN-sim. The detailed algorithm is stated in Algorithm 7.

In summary, we introduce Path Graph Neural Networks (PathGNN) to improve the generalizability of GNNs for distance related problems by improving the algorithm alignment [5]. It is a specially designed GNN layer that imitates one iteration of the classical Bellman-Ford algorithm. There are three variants of PathGNN, i.e. MPNN-Max, PathGNN, and PathGNN-sim, each of which corresponds to different degrees of flexibilities. In our experiments, they perform much better than GCN and GAT for all path-related tasks regarding the generalizability, as stated in Section 5 in the main body.

## C.3 Homogeneous prior

As described in the main body, the approach to build HomoGNN is simple: remove all the bias terms in the multi-layer perceptron (MLP) used by ordinary GNNs, so that all affine transformations degenerate to linear transformations. Additionally, only activation functions that are homogeneous are allowed to be used. Applying this approach to GN-blocks, we have the homogeneous GN blocks. The HomoMLP is defined as MLPs without biases and with homogeneous functions as the activation functions. Note that ReLU and Leaky ReLU are both homogeneous functions. The sum/max/mean poolings are also homogeneous functions.

The only non-homogeneous pooling module that is widely used in GNNs is the attentional pooling module [9, 10]. We also utilizes it as a flexible aggregation module in PathGNN as described in Section C.2. In this subsection, we present another simple approach, which is generally similar to the previous one, to design the homogeneous attentional poolings: replace MLPs with HomoMLPs and apply one normalization layer before softmax. In detail, most attentional poolings have similar architecture to the one in PathGNNs as presented in Algorithm 6. The attentional pooling modules output the weighted summation of updated features $\tilde{\mathbf{e}}_k$, where the weights $\alpha_k$ are probabilities calculated by softmax based on the scores $score_k$. The updates features $\tilde{\mathbf{e}}_k$ and scores $score_k$ are both calculated by applying some MLPs on the input features, as shown in Algorithm5. We want to design attentional poolings that are homogeneous functions over the input features. The approach is as follows: We change all MLPs to HomoMLPs and replace softmax with a scale-invariant version of softmax. In this case, the weights $\alpha_k$ will not change with respect to the scales of input features. The magnitudes of the final output of attentional pooling modules $\bar{\mathbf{e}}$, which is the weighted summation of updated features $\tilde{\mathbf{e}}_k$, will then scale linearly with respect to the magnitudes of updates features as well as the magnitudes of input features. Therefore, the resulting attentional module is a homogeneous function. To design a scale-invariant softmax, we simply adopt one normalization layer before softmax so that the effect of scales is eliminated. Typically, we build our scale-invariant softmax as

$$\text{scale-invariant-softmax}(score) = \text{softmax}\left(\frac{score}{\max score - \min score}\right) \tag{30}$$

(a) Classical GNNs / WL-test can't distinguish two circles and one circle

(b) Classical GNNs / WL-test can't distinguish regular graphs with same degrees

(c) GNNs with unique node identifications (such as random initialization) can distinguish regular graphs with same degrees

(d) Datasets visualization for distinguishing regular graphs by GNNs. Each isomorphic graph corresponds to a set of graphs with random node attributes and the problem reduce to graph classification problems.

Figure 1: Random initialization improves the representation power of GNNs by distinguishing different nodes. (a) and (b) illustrate that classical GNNs which are as powerful as WL-test for graph isomorphic problems can not distinguish regular graphs with same degrees. Typically, (a) shows an example of two graphs with different component numbers and (b) shows an example of two graphs with different diameters. Therefore, classical GNNs with constant initialization are not powerful enough to solve many graph-related problems such as component counting and graph diameters. (c) illustrates that GNNs with random initialization can distinguish regular graphs since each node is assigned with a unique identification. (d) further demonstrates the details about datasets for training and evaluating GNNs with random initialization. Specifically, GNNs can map each isomorphic graph into a set of vectors in the graph-level embedding space and then the graph-isomorphic problem reduces to the classical classification problem.

No division will be performed if $\max score = \min score$. Together with the bias-invariance property of softmax (Proposition 2 in [11]), we have for all $\lambda > 0$,

$$\text{scale-invariant-softmax}(\lambda \cdot score) = \text{scale-invariant-softmax}(score) \qquad (31)$$

Similar intuitions can be applied to design attentional poolings with activation functions other than softmax, such as sparsemax [11].

### C.4 Random Initialization

As analyzed in previous works [1, 12], standard GNNs are at most as powerful as the Weisfeiler-Leman test (WL-test) [13] for distinguishing non-isomorphic graphs. Standard GNNs are then theoretically short of representation powers for solving graph problems such as component counting and graph radius/diameters, as illustrated in Figure 1. WL-test can't distinguish the graph as one circle with 6 nodes and the graph as two circles each with 3 nodes.

However, previous works all assumed that the node attributes were initialized as constant while analyzing GNNs' representation powers. With the constant initialization, nodes with the same subtree patterns are not distinguishable similarly to the WL-test as shown in Figure 1. Nevertheless, they didn't fully utilize the representation powers of neural networks. Instead of utilizing the constant initialization techniques, the node attributes could be initialized as random numbers so that each node has its own unique identification. Subtrees that are of the same pattern but are composed with different nodes are therefore distinguishable. In this case, each graph structure $E$ is mapped into a set of input graphs $G_{random} = \{V_{random}, E\}$ where $V_{random} = \{\mathbf{v}_i = rand(), i = 1, 2, \cdots, N_v\}$. And the graph-isomorphic problem is then formulated as distinguishing sets of graphs. As illustrated in Figure 1, and also as verified in Section E.4, GNNs with random initialization can distinguish non-isomorphic regular graphs which are unable to be distinguished by WL-test. Therefore, random initialization can improve the representation powers of GNNs. [14, 15] also formulated the improvement of

representation powers by random initialization while from another perspective, specifically towards efficient universal approximators of permutation invariant/equivalent functions.

# D  Experiment Setups

We present the detailed experimental setups in this subsection. We first present details of three graph theory problems, i.e. the shortest path problem in Section D.1.2, the component counting problem in Section D.1.3, and the traveler salesman problem in Section D.1.4. We then list the experimental setups of three graph-based reasoning tasks, i.e. the physical simulation in Section D.2.1, the image-based navigation in Section D.2.2, and the symbolic Pacman in Section D.2.3. We also describe the experimental setups of graph classification in Section D.3. The descriptions of models are stated in Section D.4 and the training details are presented in Section D.5.

## D.1  Graph theory problems

We evaluate our proposals on three classical graph theory problems, i.e. shortest path, component counting, and the Traveling Salesman Problem (TSP). We present the properties of graph generators in Section D.1.1, the setups for the shortest path problem in Section D.1.2, the setups for the component counting in Section D.1.3, and the setups for the TSP problem in Section D.1.4.

### D.1.1  Properties of Graph Generators

How to sample graphs turns to be intricate in our exploration to deeply investigate the generalizability power. Four generators are adopted in our experiments:

- The Erdos-Renyi model [16], $G(n, p)$, generates graphs with $n$ nodes and each pair of nodes is connected with probability $p$.
- The KNN model, $\text{KNN}(n, d, k)$, first generates $n$ nodes whose positions are uniformly sampled from a $d$-dimensional unit cube. The nodes are then connected to their $k$ nearest neighbors. The edge directions are from the center node to its neighborhoods.
- The planar model, $\text{PL}(n, d)$, first generates $n$ nodes whose positions are uniformly sampled from a $d$-dimensional unit cube. Delaunary triangulations are then computed [17] and nodes in the same triangulation are connected to each other.
- The lobster model, $\text{Lob}(n, p_1, p_2)$, first generates a line with $n$ nodes. $n_1 \sim \mathcal{B}(n, p_1)$ nodes are then generated as first-level leaf nodes, where $\mathcal{B}$ denotes the binomial distribution. Each leaf node uniformly selects one node in the line as the parent. Afterwards, $n_2 \sim \mathcal{B}(n_1, p_2)$ are generated as second-level leaf nodes. Each second-level leaf node also uniformly selects one first-level leaf node as the parent. The parents and children are connected to each other and the graph is therefore undirected.

For the Erdos-Renyi model, we assign $p$ equal to 0.5 so that all graphs of $n$ nodes can be generated with equal probabilities. However, the expectation of graph diameters decreases dramatically as graph sizes increase for such model. As illustrated in Figure 2, the graph diameters are just 2 with high probability when the node number is larger than 50.

The other three graph generators are therefore designed to generate graphs with larger diameters for better evaluation of models' generalizabilities w.r.t. scales. We manually select their hyper-parameters to efficiently generate graphs of diameters as large as possible. Specifically, we set $d = 1$ and $k = 8$ for the KNN model, $d = 2$ for the planar model, $p_1 = 0.2$ and $p_2 = 0.2$ for the lobster model. Their properties are illustrated in Figure 2. Note that the distances are positively related to the graph sizes for all three graph generators. Moreover, for the lobster model, the distances increase almost linearly with respect to the graph sizes.

### D.1.2  Shortest Path

In this problem, given a source node and a target node, the model needs to predict the length of the shortest path between them. The edge weights are positive and uniformly sampled. We consider both unweighted graphs and weighted graphs (edge weights uniformly sampled between 0.5 and 1.5). Groundtruth is calculated by Dijkstra's algorithm.

Figure 2: Properties of different random graph generators. The upper row illustrates graph samples generated by the corresponding generator. The lower row demonstrates the relationships between the graph sizes (i.e. node numbers) and the distributions of random node pairs' distances. Typically, for each generator and each graph size, 1000 sample graphs are generated by the corresponding generator for estimating the distributions. Box plots are utilized for visualizing the distributions.

We utilize a three-dimensional one-hot representation to encode the location of the source and target nodes ($100$ for source, $010$ for destination, and $001$ for other nodes). Edge weights are encoded as the edge attributes. Two metrics are used to measure the performance of GNNs. The relative loss is first applied to measure the performance of predicting shortest path lengths. To further examine the models' ability in tracing the shortest path, we implement a simple post-processing method leveraging the noisy approximation of path lengths (described later). We define the relative loss as $|l - l_{pred}|/l$ and the success rate of identifying the shortest path after post-processing as $\mathbb{1}(l = l_{post\text{-}pred})$, where $l$ is the true shortest path length, $l_{pred}$ is the predicted length by GNNs, and $l_{post\text{-}pred}$ is the length of the predicted path after post-processing.

**Post-Processing**    After training, our model can predict the shortest path length from source to target nodes. The post-processing method is then applied to find the shortest path based on the learned models. Specifically, the post-processing method predicts the shortest path

$$p = [p_1, p_2, \cdots, p_n] = \text{post-processing}(G; GNN)$$

given the input graph $G = (V_{i,j}, E)$, as defined in Section 5.1.1., and the noisy shortest path length predicting model $GNN$, where $i, j$ are the indexes of source and target nodes, $V_{i,j}$ is the node attributes with one-hot encodings, and $p_k$ denotes the index of the $k$th node on the shortest path.

The post-processing algorithm is stated in Algorithm.8. Concretely, we first denote the shortest path length between any two nodes $i'$ and $j'$ predicted by the trained model $GNN$ as $dist_{i',j'} = GNN(G_{i',j'}) = GNN((V_{i',j'}, E))$, where $V_{i',j'}$ represents the one-hot node attributes when nodes $i'$ and $j'$ are the source and target nodes. For further convenience, we also defined $w_{i',j'}$ as the weight of edge connecting node $i'$ and node $j'$ ($w_{i',j'} = \infty$ if node $i'$ and node $j'$ are not connected by an edge.). Then, the post-processing algorithm sequentially predicts the next node $p_{k+1}$ of node $p_k$ by minimizing the difference between the predicted shortest path length approximated by GNNs $dist_{p_k,j}$ and the length of shortest path predicted by the post-processing method $dist_{p_{k+1},j} + w_{p_k,p_{k+1}}$. To further reduce the effect of models' noises, another constrain is added as $dist_{p_{k+1},j} + w_{p_k,p_{k+1}} \leq dist_{p_k,j}$ so that the method will always convergence.

### D.1.3   Component Counting

In the component counting problem, the model counts the number of connected components of an undirected graph. To generate a graph with multiple components, we first sample a random integer $m$ between 1 to 6 as the number of components, and then divide the nodes into $m$ parts. In detail, for $n$ nodes and $m$ components, we first uniformly sample $m - 1$ positions from 1 to $n - 1$ and then divide the $n$ nodes into $m$ parts by the $m - 1$ positions. We then connect nodes in each component using the random graph generator defined in Section D.1.1, e.g., the Erdos-Renyi graph and the lobster graph. The metric is the accuracy of correct counting. We initialize the node attributes by

**Algorithm 8:** Post-processing to predict the shortest path

**Input:** graph $G = (V_{i,j}, E)$; source node index $i$; target node index $j$; trained models $GNN$ that predict the shortest path length from nodes $i'$ to node $j'$ as $dist_{i',j'} = GNN((V_{i',j'}, E))$
**Output:** shortest path $p = [p_1, p_2, \cdots, p_n]$

Initialize $p_1 = i, k = 1$.
**while** $p_k \neq j$ **do**
  **if** $|\{l : dist_{l,j} + w_{p_k,l} \leq dist_{p_k,j}\}| > 0$ **then**
    $p_{k+1} = \arg\min_{l:dist_{l,j}+w_{p_k,l} \leq dist_{p_k,j}} |dist_{l,j} + w_{p_k,l} - dist_{p_k,j}|$.
  **else**
    return $p = []$
  **end if**
  $k = k + 1$.
**end while**
return p

(a) Newton's ball  (b) Symbolic PacMan  (c) Image-based Navigation

Figure 3: Figure (a) shows a set of Newton's balls in the physical simulator. The yellow arrow shows the moving direction of the first ball. Figure (b) is a scene in our symbolic PacMan environment. Figure (c) illustrates our image-based navigation task in a RPG-game environment.

random values$\in [0, 1)$, so that GNNs are powerful enough solve the component counting problem, as discussed in Section C.4.

### D.1.4 Traveler Salesman Problem (TSP)

In the Euclidean travelling salesman problem (TSP), there are several 2D points located in the Euclidean plane, and the model generates a shortest route to visit each point. The graph is complete. The weight of an edge is the Euclidean distance between the two ends. Points $\{(x_i, y_i)\}$ are uniformly sampled from $\{x, y \in \mathbb{Z} : 1 \leq x, y \leq 1000\}$. We use the standard solver for TSP, Concorde [18], to calculate the ground truth. The node attributes are the 2D coordinates of each node. We use relative loss defined the same as the shortest path problem to evaluate the networks.

### D.2 Graph-based reasoning tasks

We further evaluate the benefits of our proposals using three graph-based reasoning tasks. We describe the setups of physical simulation in Section D.2.1, the setups of symbolic Pacman in Section D.2.3, and the setups of image-based navigation in Section D.2.2.

### D.2.1 Physical Simulation

We evaluate the generalizability of our models by predicting the moving patterns between objects in a physical simulator. We consider an environment similar to *Newton's cradle*, also known as called *Newton's ball*, as shown in Figure 3(a): all balls with the same mass lie on a friction-free pathway. With the ball at one end moving towards others, our model needs to predict the motion of the balls of both ends at the next time step. The probability is 50% for balls to collide. We represent the environment as a chain graph. The nodes of the graph stand for the balls and the edges of the graph

stand for the interactions between two adjacent balls. We fix the number of balls within $[4, 34)$ at the training phase, while test the networks in environments with 100 nodes.

In detail, we generate samples as follows:

- $n - 1$ balls with same properties are placed as a chain in the one-dimensional space where each ball touches its neighbourhoods.
- A new ball moves towards the $n - 1$ balls from left position $x$ with constant speed $v$.
- The model needs to predict each ball's position and speed after one time step.

The radius of balls, $r$, is set to 0.1 and the positions are normalized so that the origin is in the middle of $n - 1$ balls. The left ball's position $x$ is set so that its distance to the most left ball among the other $n - 1$ balls is uniformly sampled between 0 and $200r$. The left ball's speed $v$ is uniformly sampled between 0 and $200r$. Note that the left ball may or may not collide with the other balls depending on its positions and weights. The probability is 50% for balls to collide.

Since the $n - 2$ balls in the middle will not change their positions or speeds in any cases, we simplify the output to the positions and speeds of left and right balls. To avoid trivial solutions, we still force the models to predict positions and speeds at the node level, which means no global readout modules are allowable.

### D.2.2 Image-based Navigation

We show benefits of the differentiability of a generalizable reasoning module using the image-based navigation task, as illustrated in Figure 3(c). The model needs to plan the shortest route from source to target on 2D images with obstacles. However, the properties of obstacles are not given as a prior and the model must discover them based on image patterns during training.

We simplify the task by defining each pixel as obstacles merely according to its own pixel values. Specially, we assign random heights from $[0, 1)$ to pixels in 2D images. The agent can't go through pixels with heights larger than 0.8 during navigation. The cases where no path exists between the source node and the target node are abandoned. We represent the image by a grid graph. Each node corresponds to a pixel. The edge attributes are set to one. The node attributes are the concatenation of pixel values and the one-hot embedding of node's categories (source/target/others). Note that, for more complex tasks, the node attributes can also include features extracted by CNNs.

In detail, we generate samples as follows:

- $n \times n$ grid is first generated. Each node is connected to its left, right, up, and bottom neighbourhoods. (The boundary situations are omitted for simplicity)
- The height, $h_i$, is uniformly sampled between 0 and 1, and is then assigned to node $i$. Nodes with heights larger than 0.8 can't be visited.
- The source node and the target node are uniformly sampled from node pairs that have height less than 0.8 and are connected.

The node attributes are their heights and the one-hot encodings of their categories (i.e. source, target, or others). The edge attributes are all ones. The properties of datasets are visualized in Figure 4.

### D.2.3 Symbolic Pacman

To show that our iterative module can improve reinforcement learning, we construct a symbolic PacMan environment with similar rules to the PacMan in Atari [19]. As shown in Figure 3(b), the environment contains a map with dots and walls. The agent starts at one position and at each step it can move to one of neighboring grids. The agent will receive reward of 1 when it reaches one dot and "eats" the dot. The discount factor is set to 0.9. The agent needs to figure out a policy to quickly "eat" all dots while avoiding walls on the map to maximize the return.

In detail, we generate random environments as follows:

- Maze of size $16 \times 16$ is first generated.
- $n_w$ walls of length 3 are then generated.

Figure 4: Illustration of the properties of the datasets for image-based navigation. Box plots are utilized to visualize the distributions of the lengths of the shortest path between two random nodes for each map-size. The shortest path lengths increase with the map sizes.

- The walls' directions are assigned randomly as vertical or horizontal.

- The walls' positions are uniformly sampled from all feasible ones regardless of overlappings.

- $n_d$ dots are further generated with positions uniformly sampled from positions that are not occupied by walls.

- one agent is at last generated with positions uniformly sampled from non-occupied ones.

The model then controls the agent to navigate in the maze to eat all dots. The action space is [left, right, up, down]. The agent will not move if the action is infeasible such as colliding with the walls. The game will stop if the agent has eaten all dots or has exceeded the maximum time step, which is $16 \times 16 \times (n_d + 1)$. A new environment will be generated afterwards. The metric is then the success rates of eating dots:

$$\frac{\text{number of eaten dots}}{\text{number of reachable dots}}$$

At each step, landmarks are placed on each corner of the shortest paths from the agent to dots. The input state is then a graph with the agent, dots, and landmarks as nodes. The node attributes are their positions plus one-hot encodings of their categories. The positions are normalized so that the agent is at the origin. The edge weights are set as the Manhattan distance between every two nodes.

We train our models using the double DQN [20] with value networks replaced by our backbones. The reward for eating each dot is 1 and no penalty for colliding with walls. The discount is set to 0.9 for each time step to encourage faster navigation. The exploration probability is 0.1 and the warm-up exploration steps are 1000. Value networks are trained every 4 time steps and updated every 200 time steps. The size of replay buffer is 10000. The batch size is 32 and the learning rate is 0.0002. Models are tested on 200 different environments and the averaged performance is reported.

The network architectures are as follows: 2-layer MLP with leaky ReLU for feature embedding, GNN/CNN modules for message passing, max pooling for readout, and 1-layer FC for predicting the Q values. The hidden sizes are 64. We compare our IterGNNs with PointNet [21], GCN [22], and CNNs. For the PointNet model, the GNN modules are identities by definition. For the GCN and CNN models, we compare the performance of their 1/3/5/7/9-layer variants and report the best of them. We also compare the choice of the kernel sizes of CNNs among 3/5/7 and report the best of them.

### D.3 Graph Classification

Note that the abilities of models to utilize information of the long-term relationships are necessary for accurately solving most of the previous tasks and problems. Therefore, the benefits of adaptive and unbounded depths introduced by our iterative module are distinguished. In this sub-section, we show that our IterGNN can also achieve competitive performance on graph-classification benchmarks, demonstrating that our iterative module does not hurt the standard generalizability of GNNs while improving their generalizability w.r.t. graph scales. The results are presented in Section E.3.

In detaile, we evaluate models on five small datasets, which are two social network datasets (IMDB-B and IMDB-M) and three bioinformatics datasets (MUTAG, PROTEINS and PTC). Readers are referred to [1] for more descriptions of the properties of datasets. We adopt the same evaluation method and metrics as previous state-of-art [1], such as 10-fold cross validation. The dataset splitting strategy and pre-processing methods are all identical to [1] by directly integrating their public codes[1].

Regarding the models, we adopt the previous state-of-art, GIN [1] as GN-blocks and the JK connections [23] plus average/max pooling as the readout module. To integrate the iterative architecture, we wrap each GN-block in original backbones with the iterative module with maximum iteration number equal to 10. The tunable hyper-parameters include the number of IterGNNs, the epoch numbers, and whether or not utilizing the random initialization of node attributes.

### D.4 Models and Baselines

We follow the common practice of designing GNN models as presented in Section F.2. We utilize a 2-layer MLP for node attribute embedding and use a 1-layer MLP for prediction. The max/sum poolings are adopted as readout functions to summarize information of a graph into one vector.

To build the core GNN module, we need to specify three properties of GNNs: the GNN layers, the paradigms to compose GN-blocks, and the prior, as stated in Section F.2. In our experiments, we explore the following options for each property:

- GNN layers: PathGNN layers and two baselines that are GCN [22] and GAT [24].
- Prior: whether or not apply the homogeneous prior
- Paradigm to compose GN-blocks: our iterative module; the simplest paradigm that stacks multiple GNN layers sequentially; the ACT algorithm [25]; and the fixed-depth shared-weights paradigm, as described in the main body.

For the homogeneous prior, we apply the prior to the node-wise embedding module, the readout module, and the final prediction module as well for most problems and tasks. However, for problems whose solutions are not homogeneous (e.g. component counting), we only apply the homogeneous prior to the core GNN module.

In detail, models are specified by the choices of the previous properties. The corresponding models and their short names are as follows:

- *GCN, GAT*: Models utilizing the multi-layer architecture (i.e. stacking multiple GN-blocks) with GCN [22] or GAT [24] as GN-blocks. The homogeneous prior is not applied.
- *Path / Multi-Path*: Models utilizing the multi-layer architecture and adopting PathGNN as defined in Section C.2 as GN-blocks. No homogeneous prior is applied.
- *Homo-Path / Multi-Homo-Path*: Models utilizing the multi-layer architecture and adopting PathGNN as GN-blocks. And the homogeneous prior is applied on all modules unless otherwise specified.
- *Iter-Homo-Path / Iter-HP*: Models utilizing the iterative module as described in Section C.1.2 and adopting PathGNN as GN-blocks. The homogeneous prior is applied on all modules unless otherwise specified.
- *Shared-Homo-Path, ACT-Homo-Path*: Models utilizing the fixed-depth and shared-weights paradigm (i.e. repeating one GN-block for multiple times) and the adaptive computation time algorithm [25], respectively. PathGNN and the homogeneous prior are all applied.

- *Iter-Path*: Same as Iter-Homo-Path except that no homogeneous prior is applied.
- *Iter-GAT*: Models utilizing our iterative module and adopting GAT as GN-blocks. No homogeneous prior is applied.

For most problems, max pooling is utilized as the readout function and we use only one IterGNN to build the core graph neural networks. The homogeneous prior is applied to all modules. However, for component counting, sum pooling is utilized as the readout function and two IterGNNs are stacked sequentially, since two iteration loops are usually required for component-counting algorithms (one for component assignment and one for counting). We utilize the node-wise IterGNN to support unconnected graphs as introduced in Section C.1.1. The homogeneous prior is only applied to the GNN modules but not the embedding module and count prediction module, because the problem is not homogeneous. Random initialization of node attributes is also applied to improve GNNs' representation powers as analyzed in Section C.4.

## D.5 Training Details

All models are trained with the same set of hyper-parameters: the learning rate is 0.001, and the batch size is 32. We use Adam as the optimizer. The hidden neuron number is 64. For models using the iterative module or the ACT algorithm, we train the networks with 30 maximal iterations and test them with no additional constraints. For the fixed-depth shared-weights paradigm, we train the networks with 30 iterations and test them with 1000 iterations (maximum node numbers in the datasets). The only two tunable hyper-parameter within our proposals are the epoch number=20, $40, \cdots, 200$ and the degree of flexibilities of PathGNN, each corresponding to one variation of the PathGNN layer as described in Section C.2. Another hyper-parameter within the ACT algorithm [25] is $\tau = 0, 0.1, 0.01, 0.001$. We utilize the validation dataset to select the best hyper-parameter and report its performance on the test datasets.

# E  Experimental results

We present experimental results that are omitted in the main body due to the space limitation in Section E.1.1. We then analyze the iteration numbers learned by ACT and our models in Section E.1.2. In Section E.2, we present the generalization performances of IterGNN, GCN and PointNet for the symbolic Pacman task in environments with different number of dots and of walls.

Other than those generalization performance w.r.t. scales, we evaluate the standard generalizability of our models on five graph classification benchmarks. As shown in Section E.3, our Iter-GIN achieves competitive performance to the state-of-art GIN [1]. We also verifies the claim in Section C.4 stating that regular graphs, which can not be distinguished by WL-test, can be distinguished by IterGNNs plus random initialization of node attributes, in Section E.4.

## E.1  Solving graph theory problems

### E.1.1  Generalize w.r.t. graph sizes

We provide the omitted generalization performance of models on the weighted shortest path problem with PL and KNN as generators in Table 1. Each of our proposals help improve the generalizability of models with respect to graph sizes and graph diameters. Our final Iter-Homo-Path model largely outperforms the other models regarding the generalizability w.r.t. scales.

### E.1.2  The iteration numbers learned by ACT and our iterative module

We first analyze the ACT [25] algorithm and show that it is easy for ACT to learn small iteration numbers. We then explain why the minimum depth of GNNs for accurately predicting the lengths $l$ of the shortest path is $l/2$ to show that our Iter-Homo-Path model actually learns an optimal and interpretable stopping criterion. At last, we evaluate the benefits of the decaying mechanism of our iterative module as described in Section C.1.2.

**The iteration numbers learned by ACT are usually small.** As shown in Figure 5, the ACT-Homo-Path model learns much smaller iteration numbers than other models. It is because

Table 1: Generalization performance on graph algorithm learning and graph-related reasoning. Models are trained on graphs of sizes within $[4, 34)$ and are tested on graph of larger sizes such as 100 (for the shortest path problem and the TSP problem) and 500 (for the component counting problem). The metric for the shortest path problem and the TSP problem is the relative loss. The metric for the component counting problem is the accuracy.

| | Shortest Path - weighted | | | | Component Cnt. | | TSP |
|---|---|---|---|---|---|---|---|
| Models | ER | PL | KNN | Lob | ER | Lob | 2D |
| GCN [22] | 0.1937 | 0.202 | 0.44 | 0.44 | 0.0% | 0.0% | 0.52 |
| GAT [24] | 0.1731 | 0.127 | 0.26 | 0.28 | 24.4 % | 0.0% | 0.18 |
| Path (ours) | 0.0014 | 0.084 | 0.16 | 0.29 | 82.3% | 77.2% | 0.16 |
| Homo-Path (ours) | **0.0008** | 0.015 | 0.07 | 0.27 | **91.9%** | 83.9% | 0.14 |
| Iter-Homo-Path (ours) | **0.0007** | **0.003** | **0.03** | **0.02** | 86.6% | **95.4%** | **0.11** |

of the formulations of the ACT algorithm. In detail, the ACT algorithm considers a different random process from our iterative module while designing the stopping criterion. The expected output has the form as $\sum_{i=1}^{k-1} c^i h^i + (1 - \sum_{i=1}^{k-1} c^i) h^k$. The notations are the same as our iterative module described in the main body. And the stopping criterion is $\sum_{i=1}^{k} c^i > 1$. Compared with the expected output $\sum_{j=1}^{k} c^j h^j \prod_{i=1}^{j-1} (1 - c^i)$ and the stopping criterion $\prod_{i=1}^{k} (1 - c^i) < \epsilon$ of our iterative module, it is generally easier for the stopping criterion of ACT to be satisfied since $1 - \sum_{i=1}^{k} c^i < \prod_{i=1}^{k} (1 - c^i)$ if $k > 1$ and $0 < c^i < 1$ for all $i <= k$. In the paper that proposes the ACT algorithm [25], the author also states similar intuitions that the formulation in our iterative module will not stop in a few steps. In fact, as far as we know, all works [25, 26, 27] that utilize the ACT algorithm adopt a noisy regularization term to encourage fewer iteration numbers. They have distinct motivations and objectives from our work. For example, most of them are designed for more efficiencies by fewer iterations [26, 27]. On the other hand, our iterative module is designed to improve the generalizability of GNNs with respect to scales by generalizing to much larger iterations.

**The minimum depth of GNNs to accurately solve the unweighted shortest path problem.** The minimum depth of GNNs to accurately predict the shortest path whose length is $l$ is $l/2$. The reason is that, due to the message-passing nature of GN-blocks, $l$-layer GNNs can at most summarize information of the shortest paths whose lengths are smaller than $2l + 1$. Therefore, if models can't iterate for distance/2 times, they can't collect enough information for making an accurate prediction of the shortest path lengths, but can only guess based on the graph's global properties (e.g. the number of nodes and edges) instead. As illustrated in Figure 5, our model, Iter+homo+decay, i.e. the Iter-Homo-Path model, learns the optimal stopping criterion, whose iteration numbers are equal to half of the shortest path length. In other words, it achieves the theoretical lower bound of the iteration numbers for accurate predictions given the message-passing nature of GN-blocks.

**The benefits of the decaying mechanism of our iterative module.** Let's compare the performance of IterGNN+homo+decay and IterGNN+homo in Figure 5, to verify the benefits of the decaying mechanism. Although IterGNN+homo is still able to generalize to the number of iterations as large as 100, it can not generalize to much larger iteration numbers such as 200 or 2000 without the decaying mechanism. Models with fewer iteration numbers than the lower bound, distance/2, theoretically lack powers for accurately predicting the shortest paths of lengths larger than $l$. The success rate of IterGNN+homo is 67.5%, which is much smaller than the success rate of IterGNN+homo+decay 100%, for predicting the shortest path on lobster graphs of size $500$. The worse performance of IterGNN+homo than IterGNN+homo+decay suggests the effectiveness of our decaying mechanism for improving the generalizability of models with respect to graph scales.

## E.2 Symbolic Pacman

The experimental setups are presented in Section D.2.3. Note that, unlike the original Atari PacMan environment, our environment is more challenging because we randomly sample the layout of maps for each episode and we test models in environments with different numbers of dots and walls. The agent can not just remember one policy to get successful but needs to learn to do planning according to the current observation.

Figure 5: The iteration numbers of GNN layers w.r.t. the distances from the source node to the target node for the unweighted shortest path problem. All of them utilize Homo-Path as the backbone and change the paradigm to compose GN-blocks, except for the "IterGNN+decay" model. Multi denotes the simplest paradigm that stacks GN-blocks sequentially. The iteration numbers for the ACT algorithm and for the IterGNN models are all adaptive to the inputs and the stopping criterions are leanred during training. Models are trained on graphs of sizes $[4, 34)$ while tested on graphs of diameters 500. The theoretical lower bound of iteration numbers for accurate prediction, i.e. distance/2, is also plotted.

Table 2: Generalization performance of IterGNN for symbolic Pacman. Metric is the success rate of eating dots. Models are trained in environments with 10 dots and 8 walls and are tested in environments with different number of walls and dots.

| #wall \ #dots | 1 | 5 | 10 | 15 | 20 |
|---|---|---|---|---|---|
| 0 | 1.00 | 1.00 | 0.99 | 0.99 | 1.00 |
| 3 | 0.95 | 1.00 | 0.98 | 0.94 | 0.98 |
| 6 | 0.90 | 0.95 | 0.94 | 0.97 | 0.98 |
| 9 | 0.80 | 0.92 | 0.95 | 0.95 | 0.93 |
| 12 | 0.60 | 0.96 | 0.97 | 0.98 | 0.93 |
| 15 | 0.75 | 0.92 | 0.94 | 0.95 | 0.97 |

Table 2, Table 3, and Table 4 show the performance of IterGNN, GCN [22] and PointNet [21], respectively, in environments with different number of walls and dots. Our IterGNN demonstrates remarkable generalizability among different environment settings, as stated in Table 2. It successfully transfers policies to environments with different number of dots and different number of walls. IterGNN performs much better than the GCN and PointNet baselines, demonstrating that our proposals improve the generalizability of models. GCN performs the worst probably because of the unsuitable strong inductive bias encoded by the normed-mean aggregation module.

### E.3 Graph Classification

At last, we evaluate models on standard graph classification benchmarks to show that our proposals do not hurt the standard generalizability of GNNs while improving their generalizability w.r.t. scales. More descriptions of the task are available in [1]. The experimental setups are presented in Section D.3.

As stated in Table 5, our model performs competitively with the previous state-of-art backbone, GIN, on all five benchmarks. It suggests that our iterative module is a safe choice for improving generalizability w.r.t. scales while still maintaining the performance for normal tasks. Note that, due to the shortage of time, little hyper-parameter search was conducted in our experiments. Default

Table 3: Generalization performance of GCN for symbolic Pacman. Metric is the success rate of eating dots. Models are trained in environments with 10 dots and 8 walls and are tested in environments with different number of walls and dots.

| #dots / #wall | 1 | 5 | 10 | 15 | 20 |
|---|---|---|---|---|---|
| 0 | 0.05 | 0.23 | 0.09 | 0.07 | 0.05 |
| 3 | 0.10 | 0.23 | 0.20 | 0.15 | 0.04 |
| 6 | 0.00 | 0.32 | 0.17 | 0.09 | 0.06 |
| 9 | 0.20 | 0.27 | 0.23 | 0.06 | 0.10 |
| 12 | 0.05 | 0.18 | 0.26 | 0.13 | 0.03 |
| 15 | 0.05 | 0.17 | 0.23 | 0.15 | 0.08 |

Table 4: Generalization performance of PointNet for symbolic Pacman. Metric is the success rate of eating dots. Models are trained in environments with 10 dots and 8 walls and are tested in environments with different number of walls and dots.

| #dots / #wall | 1 | 3 | 6 | 9 | 12 | 15 |
|---|---|---|---|---|---|---|
| 2 | 0.82 | 0.58 | 0.39 | 0.32 | 0.34 | 0.29 |
| 4 | 0.72 | 0.48 | 0.31 | 0.25 | 0.24 | 0.22 |
| 6 | 0.71 | 0.31 | 0.36 | 0.24 | 0.19 | 0.14 |
| 8 | 0.60 | 0.36 | 0.21 | 0.20 | 0.20 | 0.28 |
| 10 | 0.50 | 0.33 | 0.33 | 0.29 | 0.16 | 0.15 |

hyper-parameters such as learning rate equal to 0.001 and hidden size equal to 64 were adopted. Therefore, the performance of Iter-GIN is potentially better than those stated in Table 5.

## E.4 Effectiveness of Random Initialization

As discussed in Section C.4, we adopt random initialization to improve GNNs' representation powers especially for solving graph-related problems such as component counting and graph diameters. The argument is based on that non-isomorphic regular graphs with random initialized node attributes are distinguishable by GNNs, i.e. GNNs can distinguish sets of graphs as illustrated in Figure 1(d). Although it has been proved theoretically in the bounded settings [14, 15], we further verify its effectiveness in practice.

The task is then as simple as a binary classification problem to distinguish regular graphs as shown in Figure 1(a) and Figure 1(b). 10000 samples are generated for training with graph structures uniformly sampled from two regular graphs and the node attributes uniformly sampled between 0 and 1. One thousand samples are then generated randomly for test. No validation set is needed as we do not perform hyper-parameter tuning. Experimental results show that our Iter-HomoPath model achieves 100% accuracy for distinguishing both pairs of non-isomorphic regular graph.

| Dataset | GIN | Iter-GIN |
|---|---|---|
| IMDB-B | 75.1±5.1 | **75.7±4.2** |
| IMDB-M | **52.3±2.8** | 51.8±4.0 |
| MUTAG | 89.4±5.6 | **89.6±8.6** |
| PROTEINS | 76.2±2.8 | **76.3±3.4** |
| PTC | **64.6±7.0** | 64.5±3.8 |

Table 5: The performance of our iterative module on graph classification on five popular benchmarks. Iter-GIN is built by wrapping each GIN module in the previous state-of-art method [1] using our iterative module. Metric is the averaged accuracy and STD in 10-fold cross-validation.

# F  Backgrounds - Graph Neural Networks

In this section, we briefly describe the graph neural networks (GNNs). We first present the GN blocks, which generalize many GNN layers, in Section F.1 and then present the common practice of building GNN models for graph classification/regression in Section F.2. The notations and terms are further utilized while describing PathGNN layers in Section C.2 and while describing the models/baselines in our experiments in Section D.4.

## F.1  Graph Network Blocks (GN blocks)

We briefly describe a popular framework to build GNN layers, called Graph Network blocks (GN blocks) [6]. It encompasses our PathGNN layers presented in Section C.2 as well as many state-of-art GNN layers, such as GCN [22], GAT [24], and GIN [1]. Readers are referred to [6] for more details. Note that we adopt different notations from the main body to be consistent with [6]. Also note that, even when the global attribute is utilized, the fixed-depth fixed-width GNNs still lose a significant portion of powers for solving many graph problems as proved in [28]. For example, the minimum depth of GNNs scales sub-linearly with the graph sizes for accurately verifying a set of edges as the shortest path or a s-t cut, given the message passing nature of GNNs.

The input graphs are defined as $G = (\mathbf{u}, V, E)$ with node attributes $V = \{\mathbf{v}_i, i = 1, 2, \ldots, N_v\}$, edge attributes $E = \{(\mathbf{e}_k, s_k, r_k), k = 1, 2, \ldots, N_e\}$, and the global attribute $\mathbf{u}$, where $s_k$ and $r_k$ denote the index of the sender and receiver nodes for edge $k$, $\mathbf{u}$, $\mathbf{v}_i$, and $\mathbf{e}_k$ represent the global attribute vector, the attribute vector of node $i$, and the attribute vector of edge $k$, respectively.

The GN block performs message passing using three update modules $\phi^e, \phi^v, \phi^u$ and three aggregation modules $\rho^{e \to v}, \rho^{e \to u}, \rho^{v \to u}$ as follows:

1. Node $s_k$ sends messages $\mathbf{e}'_k$ to the receiver node $r_k$, which are updated according to the related node attributes $\mathbf{v}$, edge attributes $\mathbf{e}$, and global attributes $\mathbf{u}$.

$$\mathbf{e}'_k = \phi^e(\mathbf{e}_k, \mathbf{v}_{r_k}, \mathbf{v}_{s_k}, \mathbf{u}), k = 1, 2, \ldots, N_e$$

2. For each receiver node $i$, the sent messages are aggregated using the aggregation module.

$$\bar{\mathbf{e}}'_i = \rho^{e \to v}(\{\mathbf{e}'_k | r_k = i\}$$

3. The aggregated messages are then utilized for updating the node attribute together with the related node attributes $\mathbf{v}_i$ and global attributes $\mathbf{u}$.

$$\mathbf{v}'_i = \phi^v(\bar{\mathbf{e}}'_i, \mathbf{v}_i, \mathbf{u})$$

4. The sent messages $\mathbf{e}'_k$ can also be aggregated for updating the global attribute $\mathbf{u}$ as

$$\bar{\mathbf{e}}' = \rho^{e \to u}(\{\mathbf{e}'_k, k = 1, 2, \ldots, N_e\}$$

5. The node attribute $\mathbf{v}'_i$ can be aggregated for updating the global attribute $\mathbf{u}$ as well.

$$\bar{\mathbf{v}}' = \rho^{v \to u}(\{\mathbf{v}'_i, i = 1, 2, \ldots, N_v\}$$

6. The global attribute $\mathbf{u}$ are updated according to the aggregated messages $\bar{\mathbf{e}}'$, aggregated node attributes $\bar{\mathbf{v}}'$, and previous global attribute $\mathbf{u}$.

$$\mathbf{u}' = \phi^u(\bar{\mathbf{e}}', \bar{\mathbf{v}}', \mathbf{u})$$

$\phi^e$ is often referred as the message module, $\rho^{e \to v}$ as the aggregation module, $\phi^v$ as the update module, $\rho^{e \to u}$ and $\rho^{v \to u}$ as the readout modules, in many papers. The three update modules are simple vector-to-vector modules such as multi-layer perceptrons (MLPs). The three aggregation modules, on the other hand, should be permutation-invariant functions on sets such as max pooling and attentional pooling [29, 30].

### F.2 Composing GN blocks for graph classification / regression

We follow the common practice [6, 31, 1] in the field of supervised graph classification while building models and baselines in our experiments. Typically, the models are built by sequentially stacking the node-wise embedding module, the core GNN module, the readout function, and the task-specific prediction module. The node-wise embedding module corresponds to GN blocks that are only composed of function $\phi^v$ for updating node attributes. More intuitively, it applies the same MLP module to update all node attribute vectors. The core GNN module performs message passing to update all attributes of graphs. The readout function corresponds to GN blocks that only consist of the readout modules such as $\rho^{e \rightarrow u}$ and $\rho^{v \rightarrow u}$ to summarize information of the whole graph into a fixed-dimensional vector **u**. The task-specific prediction module then utilizes the global attribute vector **u** to perform the final prediction, such as predicting the number of connected components or the Q-values within the symbolic Pacman environment.

We need to specify three properties while designing the core GNN modules: (1) the internal structure of GN blocks; (2) the composition of GN blocks; and (3) the prior that encodes the properties of the solutions of the problem.

- The internal structure of GN blocks defines the logic about how to perform one step of message passing. It is usually specified by selecting or designing the GNN layers.

- The composition of GN blocks defines the computational flow among GN-blocks. For example, the simplest paradigm is to apply multiple GN blocks sequentially. Our iterative module introduces the iterative architecture into GNNs. It applies the same GN block for multiple times. The iteration number is adaptively determined by our iterative module.

- The prior is usually specified by adopting the regularization terms. For example, regularizing the L2 norm of weights can encode the prior that GNNs representing solutions of the problem usually have weights of small magnitudes. Regularizing the L1 norm of weights can encode prior about sparsity. We can utilize the HomoMLP and HomoGNN as described in the main body to encode the homogeneous prior that the solutions of most classical graph problems are homogeneous functions.

## Footnotes

[1] https://github.com/weihua916/powerful-gnns