[Reviews · NeurIPS 2020]

Review 1

Summary and Contributions: They proposed Iterative GNN (IterGNN) and Homogeneous GNN (HomoGNN) to improve the generalizability of GNNs with respect to graph scales. The IterGNN adaptively terminates the message passing process (for graph size) and the HomoGNN converts ordinary GNN to be homogeneous (for graph weights). Empirically, they demonstrate the claimed generalization ability and their method outperforms the baselines.

Strengths: 1. The paper is well written and easy to understand. 2. The proposed IterGNN and HomoGNN successfully address the issue of ordinary GNNs they posted, and the effectiveness of the two augmentations are well supported by the experiments. 3. They brought in the homogeneous function to the graph NN, which is relatively fresh. With both theoretical and empirical results, it has been shown as an important part that enables the generalization within a set of experimental problems (including shortest path problem), and potentially other graph theory problems. 4. The experimental results are adequate and informative. The ablation studies help identify the role of proposed modules and strengthen the claims made. 5. The generalization with respect to the graph scale and weights is an interesting direction and glad to see this line of works.

Weaknesses: 1. The idea of adopting an adaptive way to determine the end of iterating is not new [31]. They modified the formulation slightly to fit the problem, but it is incremental. 2. Most experimental environments are closely related to the shortest path problem, while the design of architecture (PathGNN) includes the bias (align to Bellman-Ford) towards this kind of problem. It would be more convincing to conduct more ablation study or analysis on problems other than shortest path problems.

Correctness: The claims are well supported by both theoretical and empirical analysis.

Clarity: The paper is well written and easy to understand.

Relation to Prior Work: The previous works are almost properly discussed. (see comment below)

Reproducibility: Yes

Additional Feedback: Comments on the main text: 1. Line 78-79: "There is either no work reporting notable generalization performance with respect to graph scales." is not accurate. [r1] attempted the shortest path problem introduced in the Differentiable Neural Computer [22] and showed the generalization with respect to the size of the graph. [r2] modeled SAT problems as bipartite graphs and use message passing to tackle the problem. They observed the generalization among the size of the graph. 2. The results of the Iter-Path variant should be better included in Table 1 and Table 2 for a better understanding of the effectiveness of Iter- and Homo- augmentations. 3. Line 81: Typo "[25, 26, 26]" 4. Line 206: Typo "homogeneous functions function f" Questions: 1. The performance on component counting is high but not that close to perfect, while the flood-fill algorithm should be able to solve that and it seems that is under the representation power of your proposed model, did you investigate why it cannot solve that near perfectly? 2. Have you tried the multi-source shortest path problems (or pair-wise shortest path)? Can your model solve that efficiently as well? [UPD] I appreciate author's clarification and the auxiliary experimental results. My major concern is still whether this approach can be applied to problems other than shortest path (or similar). Overall, I would maintain my original score and vote for the acceptance of the paper. [r1] Honghua Dong*, Jiayuan Mao*, Tian Lin, Chong Wang, Lihong Li and Denny Zhou: Neural Logic Machines. In the 7th International Conference on Learning Representations (ICLR), 2019. [r2] Daniel Selsam, Matthew Lamm, Benedikt Bunz, Percy Liang, Leonardo de Moura and David L. Dill: Learning a SAT Solver from Single-Bit Supervision. In the 6th International Conference on Learning Representations (ICLR), 2018.


Review 2

Summary and Contributions: The paper targets the problem of generalization of graph neural networks (GNNs) to different graph sizes and scales. The authors suggest two main extensions of GNNs to address this issue: (1) applying a variable amount of message passing steps,that is controlled by a new learnable termination module. This makes sense since the number of iterations in many graph tasks depends on the size of the graph (2) using homogenous GNNs (by removing the biases). This makes sense since many graph tasks are homogenous with respect to the graph features (e.g. optimal value of TSP). The authors also provide theoretical analysis of the main building block - homogenous MLP (expressivity and generalization). The authors provide an extensive and convincing experimental evaluation to show that their method indeed improves performance on various tasks. From my point of view the iterative module is a significant contribution that suggests a solution to a problem that was overlooked by the graph learning community for a very long time: bounded depth GNNs have restricted expressivity. The homogenous prior is a nice but secondary contribution.

Strengths: * important problem: the paper targets the ability of GNNs to generalize to larger graphs. This is an interesting and important question. Moreover, as stated above, the iterative module is a significant contribution that suggests a solution to a problem that was overlooked by the graph learning community for a very long time: there are graph learning tasks (and some of them are very simple) that constant depth gnns cannot solve. * The paper provides an extensive evaluation section showing that the suggested extensions work well.

Weaknesses: The paper has some organization problems and I think it will benefit from some changes: (i) I believe that the iterative module is the significant contribution of this paper and should be more emphasized (ii) The theoretical analysis of the homo-mlp modules is nice but seems a bit unrelated to the flow of the paper. (iii) the path GNN module is only briefly mentioned in the main test. I would suggest removing it or doing a better job explaining it.

Correctness: Looks OK.

Clarity: The writing can be improved, See my comments in the Weaknesses section above. Nevertheless this is not a big issue and I think it can be addressed during the revision period.

Relation to Prior Work: As far as I can tell, the work is properly positioned with respect to previous work. I would add a discussion on the problem of training deep gnns which became popular recently and is related to the iterGNN architecture.

Reproducibility: Yes

Additional Feedback: Some additional comments/questions: * There is a large body of work on the problems in training/using deep GNNs (see “pairnorm” from ICLR2020 as an example). Can the authors discuss this issue with respect to their work? I would expect that applying a 100 layer gnn to a graph would not work well. * L 141: “ we can then determine the iteration numbers” - can the authors rephrase? I am not sure I understood this part. What are iteration numbers? * L. 147: the authors choose to return the expected value of h. Can the authors discuss this choice with respect to other simple choices like returning the last h. This choice seems significant as it is related to the stopping mechanism and I think that the authors can do a better job explaining it. * An illustrated example for algorithm 1 with actual “c” values might improve understanding. * Theorem 2 - can a version that uses l_infinity norm be used? * Baselines: the authors used 30 layer gnns which were shown to have problematic behavior in previous work. Can the authors add baselines with less layers (3+10 layers for example?) * If the authors use random features for graph theory problems this should be clearly stated in the main text. -------- After reading the author response and the other reviews I decided to keep my score. As I wrote in the original review I believe that the stopping mechanism is the most important contribution of this paper and I would be happy if the authors reflect that, as well as my other suggestions, in the paper.


Review 3

Summary and Contributions: summary: the authors address the problem that GNN hyperparameters depend on various aspects of graph scale (graph size, graph diameter, edge weights) and this limits the ability of a single model to generalize. they introduce two techniques to add invariance to graph scale: (1) an adaptive stopping criterion such that number of iterations can be dynamically adjusted for the given graph and (2) a layer enforcing a homogeneity prior to generalize to out of distribution edge weights. they demonstrate the performance of their model on image navigation, symbolic pacman, and graph theory problems. in each of these experiments, evaluation requires the model to generalize to graphs of substantially different sizes. contributions: theoretical: learn to terminate message passing process adaptively theoretical: graph-theoretic motivated homogenous transformation layers that appy homogeneity prior to generalize to out of distribution features and attributes theoretical: path GNN empirical: GNN shows strong performance on a variety of interesting benchmarks

Strengths: empirical evaluation is compelling: the authors compare to popular baselines and demonstrate their model in a variety of experiments. they also show ablation studies demostration that while homo-path and iter-path individually improve success, their combination provides substantial improvement.

Weaknesses: results for homo-path and iter-homo-path shown in tables 1-2: why not iter-path? would be instructive to see how this model performs to assess whether the iter or the homo-iter interaction is responsible for the performance boost. why does homogeonous prior seem to help with generalizing over graph size? the interpretation proposed was that it helps with generaliation over feature size, but not clear how this manifests in the tasks described in table 1 and 2(left) where generalization over graph size is what is being tested.

Correctness: as far as i can tell yes

Clarity: quite clear and straightforward to follow. expository figures are helpful, and motivation is clear.

Relation to Prior Work: as far as i can tell yes.

Reproducibility: Yes

Additional Feedback:


Review 4

Summary and Contributions: This paper tries to address the potential generalization problem of GNNs on graphs with variable sizes. There are three components that are proposed to address this concern: a head to control the number of steps, the homogeneous prior, and pathGNN. Experiment results show some promising improvements over two basic baseline models.

Strengths: + The paper is working towards a potentially important problem on the generalization of GNN models. + Experiment results show some improvements.

Weaknesses: Overall I feel there are too many hand-crafted elements in this full model and it's hard to know which element contributes the most. The generalizability of those elements are also concerning. Also, the experiment is designed on some non-standard benchmarks with only a few very basic GNN variants compared. In detail - For the iterative module, it really feels like this is just another use of the gates in Gated GNN or LSTM. There is no obvious benefit of using this design over the previous models (Graph ODE, Continuous GNN, etc.) where the model has a well-studied "controller" to adapt the step size and iterations for approximating the integral curves. - The pathGNN part seems to use a lot of special designs on GNNs and is bespoke for specific types of problems. It's hard to know how those special designs are gonna generalized to other problems. Also, the PathGNN needs to be moved to main text if it's describing the main and important part of the model. - There are many other GNN models that needs to be compared (e.g. Continuous GNNs, Position-aware GNNs, etc.), otherwise it's hard to tell whether the proposed model is actually making progress. - There is another dataset [1] also used TSP as a task and seems to be well benchmarked. Just as an suggestion, the authors could also consider to use that more standard dataset for comparison. [1] Benchmarking graph neural networks, Dwivedi et al.

Correctness: Yes (didn't fully check the math)

Clarity: The clarity and organization of the paper could be improved.

Relation to Prior Work: Some prior works are missing from comparison

Reproducibility: No

Additional Feedback:

[Author Response · NeurIPS 2020]

We thank the reviewers for their valuable comments. We will fix all typos and re-organize the contents in the final
nine-page paper, as advised. We will add the suggested related work, clarify the iterative module, and include more
descriptions of PathGNN and more experimental details in the main text of the final paper.

**Comparison to Iter-Path and 3/10/100-layer GNNs (R1 & R2 & R3):** A: We first compare with Iter-Path (a baseline
w.o. homogeneous prior) on more tasks. The conclusions remain unchanged. Currently, the available results are:

| Shortest Path | | Component Cnt. | | Physical sim. | | Image-based Navi. | |
|---|---|---|---|---|---|---|---|
| 0.0005 (ER) | 0.09 (Lob) | 89.2% (ER) | 89.5% (Lob) | 0.13 (50) | 5.78 (100) | 89.4% ($16 \times 16$) | 78.6% ($33 \times 33$) |

We also evaluate 3/10/100-layer PathGNNs. Their generalization performances on the shortest path (Lob) are 0.52, 0.26,
0.13, respectively, which are not as accurate as our IterGNN (0.02). We will include other results in the revised version.

**R1: Difference between our IterGNN and ACT [31].** A: Compared with ACT, several important differences enable
IterGNN to achieve much larger iteration numbers so that it can generalize to very large graphs. As illustrated in Figure
3, our IterGNN can iterate for 2500 times during inference while ACT only iterates for less than 30 times (more analysis
in Section E.1.2 and C.1.2). **Not perfect performance on component counting.** A: Ordinary GNNs need random
node features to increase representation power for solving the component counting problem, which makes optimization
difficult (due to the high variance) and causes imperfect performance. **Other shortest path problems.** A: Our model
can easily solve one of the multi-source shortest path problems, i.e., finding the distance to the nearest source nodes.
We do not know any straightforward way to find pairwise distances efficiently. **Related works reporting notable**
**generalization performance w.r.t. graph scales.** A: We will include them in the revised version. [r1] only evaluated
on graphs with limited diameters ($\leq 5$) so that their fixed-depth paradigm would work. [r2] manually set the number
of iterations as large as possible for all samples to achieve more generalizability. This approach is computationally
expensive and makes performance worse in our experiments (Shared-Homo-Path in Table 3) possibly because of the
accumulated errors after unnecessary iterations.

**R2: Reasons for not returning the last $h$ & Rephrase the iterative module with examples.** A: We cannot return
the last $h^K$ at the step $K$ because if so, the terminal signal $c^i = g(h^i)$, where $g$ is the stopping criterion function, is not
involved in the computation of $h^K = f \circ f \circ \cdots f(h^0)$ and no gradients would be propagated to train the function $g$.
Instead, we return the expected value of $h$ as $h = \sum_{i=1}^{\infty} c^k \prod_{i=1}^{k-1}(1 - c^i)h^k$ so that there are gradients to $c^i$, enabling
us to train the function $g$ end-to-end without extra supervisions. We can interpret $h$ as the expectation of the following
random process: at each step $i$, we stop the iteration and output $h^i$ with probability $c^i$. Then, starting from step 1, the
probability for the random process to stop at step $k$ and output $h^k$ is $p^k = c^k \prod_{i=1}^{k-1}(1 - c^i)$. The expectation is then
$h$. The iteration number is the number of iterations. We will include an example in the main text. **Training/using**
**deep GNNs.** A: We agree it would not work well. The layer numbers of those deep GNNs are still fixed or bounded.
Therefore, they cannot solve many simple graph-related tasks, as proved in [19]. Deep GNNs are also computationally
expensive and difficult to train. Moreover, the works that aim at training deep GNNs are orthogonal to our work. We
may incorporate those techniques, e.g. for better convergence. **Using $L_\infty$ norm for Theorem 2.** A: Yes. We provide
an arbitrary-width version of Theorem 2 (Theorem B.2.2 in Appendix) that can incorporate the $L_\infty$ norm.

**R3: Why does homogeneous prior seem to help with generalizing over graph size?** A: In general, for graphs of
larger sizes, we expect that the scale of some internal features will also be large to capture those graph properties related
to the graph size, e.g., the distances or community numbers (Detailed example in L33-40). The homogeneous prior
helps handle those out-of-range features which then improves the generalizability w.r.t. graph sizes.

**R4: Contributions and generalizability of the elements.** A: The generalization performances are shown in Table
1,2,3. We did extensive ablation studies by adding components one-by-one and by removing each part from our
best model. We fixed all other hyper-parameters for a fair comparison, and the results showed the contribution of
each element clearly. **Non-standard benchmarks with only a few very basic GNN variants compared.** A: We
have already compared our models with the most popular GNN variants. As for the benchmark, there is no such
standard benchmark evaluating the generalizability of GNNs w.r.t. scales, so we adopt common practices in related
fields when designing experiments, such as diverse graph generators for the shortest path problem and the exact same
TSP-problem generators as the benchmark paper mentioned by the reviewer R4. **Comparison with the gates in**
**LSTM/GatedGNN.** A: Although our IterGNN involves multiplications of hidden representations (like the gates), it is
significantly different from LSTM/Gated-GNNs in terms of internal structures and usages. LSTM and Gated-GNNs
require extra supervisions of the stopping conditions during training, e.g., the End-of-Sequence (EoS) symbol. **Other**
**related works.** A: Flow-based models (e.g., Graph ODE or Continuous GNNs), as stated in L99-101, do not explicitly
*learn* the iteration controller. It is unclear whether they can solve the shortest path problem on graphs of arbitrary sizes.
*By the way, Graph ODE and Continuous GNNs are not published at the submission time.* The Position-aware GNN is
proposed to improve the representation power of GNNs, similar to our random node features, instead of improving the
generalizability w.r.t. scales. **PathGNN is bespoke for specific types of problems.** A: PathGNN is not the focus and
main contribution of our work. We focus on the IterGNN and the homogeneous prior to improve the generalizability of
GNNs w.r.t. graph scales, as stated in the introduction.

[Meta-Review · NeurIPS 2020]

All reviewers read the author response and engaged actively in the discussion. During the discussion, R2 and R3 defended their positive stance while R4 stayed negative. R2 praised the stopping mechanism as an important contribution of the paper, while R3 particularly liked the empirical validation and ablation experiments. R1 found the paper incremental, but novel. I acknowledge the concerns of R4 regarding better comparison to a wider spectrum of graph neural network approaches. Like R1, I also believe it would be more compelling to test on problems less related to the shortest path problem. That said, I agree with R1, R2 and R3 that this paper makes an interesting contribution and strongly encourage the authors to work towards incorporating the feedback of all reviewers but in particular R4 and R1. The AC also discussed this paper with the senior AC, and both agree that the paper has merits that warrant publication. I am therefore recommending acceptance of the paper.